# Adaptive Online Packing-guided Search for POMDPs

**Chenyang Wu[1], Guoyu Yang[1], Zongzhang Zhang[1],[*] Yang Yu[1, 2],**
**Dong Li[3], Wulong Liu[3], Jianye Hao[3]**

[1]National Key Lab for Novel Software Technology, Nanjing University, Nanjing 210023, China
[2]Pazhou Lab, Guangzhou 510330, China
[3]Noah's Ark Lab, Huawei Company
{wucy, yanggy}@lamda.nju.edu.cn, {zzzhang, yuy}@nju.edu.cn,
{lidong106, liuwulong, haojianye}@huawei.com

## Abstract

The partially observable Markov decision process (POMDP) provides a general framework for modeling an agent's decision process with state uncertainty, and online planning plays a pivotal role in solving it. A belief is a distribution of states representing state uncertainty. Methods for large-scale POMDP problems rely on the same idea of sampling both states and observations. That is, instead of exact belief updating, a collection of sampled states is used to approximate the belief; instead of considering all possible observations, only a set of sampled observations are considered. Inspired by this, we take one step further and propose an online planning algorithm, Adaptive Online Packing-guided Search (AdaOPS), to better approximate beliefs with adaptive particle filter technique and balance estimation bias and variance by fusing similar observation branches. Theoretically, our algorithm is guaranteed to find an $\epsilon$-optimal policy with a high probability given enough planning time under some mild assumptions. We evaluate our algorithm on several tricky POMDP domains, and it outperforms the state-of-the-art in all of them. Codes are available at https://github.com/LAMDA-POMDP/AdaOPS.jl.

## 1 Introduction

POMDPs generalize the MDPs by considering the state uncertainty [1]. In MDPs, the agent knows its state exactly. In POMDPs, though an MDP determines the underlying dynamics, the agent cannot access its actual state at each timestep and, instead, receives an observation serving as a clue. In order to make decisions in POMDPs, the state uncertainty must be handled. A popular method for tackling the uncertainty is to maintain a belief, a distribution on the state space, where the probability of a state indicates how possible the agent believes it is. However, an exact belief updating requires $O(|\mathcal{S}|^2)$ computations, which is unaffordable in immense state space. A practical way is to approximate the belief with a collection of weighted particles (or samples) and update it with particle filtering [2].

Finding an optimal solution for finite-horizon POMDPs or a near-optimal solution for discounted infinite-horizon POMDPs is proved to be PSPACE-complete [3, 4]. Therefore, online planning, which computes a real-time solution, plays a crucial role in solving POMDPs, following for two reasons. The first is that online planning searches for a solution for not all possible beliefs but the current belief alone, relieving the computational load significantly. Second, online planning can take advantage of the approximate offline solution and other heuristics to expedite the tree search process.

Online planning algorithms have made substantial progress in solving large-scale POMDP problems [5–9]. These methods approximate the belief with a collection of sampled states instead of updating

---

[*]Corresponding Author

it exactly and consider a set of sampled observations instead of all possible observations. However, the estimation derived by simple belief approximation methods that have been applied, such as direct sampling and sequential importance sampling [10], have a variance increasing exponentially with searching depth. Besides, naive sampling for observations, either the direct sampling as in [7] or importance sampling as in [9], yields a large variance when the sample size is small. Inspired by this, we take one step further and propose an online planning algorithm, Adaptive Online Packing-guided Search (AdaOPS), to better approximate beliefs and better balance estimation bias and variance. It features two innovations, adaptive particle filtering, and belief packing.

The adaptive particle filter is designed for online planning. During the belief updating, it resamples adaptively and adjusts the sample size on the fly according to the dispersion of the belief distribution, which allows achieving high approximation accuracy with only a handful of particles.

AdaOPS forms a belief packing by merging similar beliefs. Only beliefs in the packing are explored, hence the name "packing-guided search". Belief packing is meant to balance estimation bias and variance. With the use of beliefs in their exact form, it is guaranteed that similar beliefs yield a slight difference in their optimal values [11]. We further extend this conclusion to cases where beliefs are approximated by a collection of weighted particles and prove that by merging similar beliefs, our algorithm can converge to the $\epsilon$-optimal policy with high probability under some mild assumptions.

## 2  Background

### 2.1  POMDPs

An MDP is mathematically defined as a tuple $(\mathcal{S}, \mathcal{A}, T, R, \gamma)$, where $\mathcal{S}$ and $\mathcal{A}$ are the state and action spaces, respectively, and $T$ and $R$ are the transition and reward functions, respectively. $\gamma \in [0, 1]$ is the discount factor. A POMDP problem can be seen as an extension to an MDP problem, which introduces two additional components, the observation space $\mathcal{O}$ and the observation function $Z$. At each timestep, instead of observing the actual state as an MDP agent does, a POMDP agent can only get an observation according to the observation function $Z(o \mid s', a) = \Pr(o \mid s', a)$, which determines the probability of observing $o \in \mathcal{O}$ after executing action $a \in \mathcal{A}$ and reaching state $s' \in \mathcal{S}$. The transition function $T$ and observation function $Z$ are sometimes hard to define explicitly. An alternative is the generative model $s', o, r \leftarrow G(s, a)$, which implicitly defines the transition and observation distributions by randomly generating a possible combination of next state $s'$, observation $o$, and reward $r$ given state $s$ and action $a$. The POMDP agent maintains a belief $b$ in states, a distribution on $\mathcal{S}$, and seeks to find a policy $\pi$ mapping beliefs to actions, which maximizes the expected sum of discounted rewards, $\mathbb{E}[\sum_{t=0}^{\infty} \gamma^t R(s_t, \pi(b_t)]$.

### 2.2  Particle Filters

Bayes filter recursively updates the posterior of the hidden states from a sequence of noisy and incomplete observations. In POMDPs, this technique is usually used to maintain a belief approximation.

Particle filter [10] is an essential class of Bayes filter, which approximates the belief with a collection of weighted particles, $\{(w_i, s_i)\}_{i \in [N]}$, where $s_i$ is the $i$-th particle, and $w_i$ is the corresponding normalized weight. Sequential importance sampling (SIS) is a widely used particle filtering method. It updates the belief by propagating particles with action $a$, $s_i' \sim T(\cdot \mid s_i, a)$, and reweighting particles with observation $o$, i.e., $w_i' = w_i Z(o \mid s_i', a)/\rho$, where $\rho$ is a normalizing factor. However, continuously updating weights will lead to sample degeneracy where most particles become impossible and have negligible weights. This problem causes the estimation variance of SIS to increase exponentially with depth [10].

In another particle filtering method, sequential importance resampling (SIR), the weight updating is followed by a resampling step, which samples $N$ particles from $\{(w_i', s_i')\}_{i \in [N]}$ with probability proportional to the weights, and the weights of the resampled particles are set to be $1/N$. Resampling alleviates the sample degeneracy by replicating high-weight particles and dropping low-weight ones. Theoretically, resampling is critical, without which we cannot derive the uniform convergence in the time horizon for particle filters [12]. However, frequent resampling induces sample impoverishment, which means that particles' diversity is diminished, and the whole collection of particles contains few distinct ones. In practice, resampling is only performed when necessary. A rule of thumb is

to resample when the ratio of particle number $N$ to effective sample size (ESS) is greater than a threshold, say two. Here, ESS is the sample size required from the true distribution to achieve the same sampling error as sampling $N$ samples from the proposal distribution and can be approximately calculated [13, 14]:

$$\text{ESS} \approx \frac{1}{\sum_{n=1}^{N} w_n^2}. \tag{1}$$

Intuitively, this strategy ensures that we only resample when the weight disparity is significant.

## 2.3 KLD-Sampling

Instead of using a fixed sample size, [15] proposes KLD-Sampling for adapting the sample size in particle filters according to the dispersion of the posterior distributions. Assuming the true posterior can be represented by a discrete piece-wise constant distribution consisting of $k$ different multidimensional bins, it calculates the number of samples needed to ensure that the Kullback-Leibler divergence (KLD) between the approximated distribution and the true distribution is less than an error $\zeta$ with a confidence level $1 - \eta$. The number of samples $N$ is given by

$$N \approx \frac{k-1}{2\zeta} \left( 1 - \frac{2}{9(k-1)} + \sqrt{\frac{2}{9(k-1)}} z_{1-\eta} \right)^3, \tag{2}$$

where $z_{1-\eta}$ is the $1 - \eta$ quantile of the standard normal distribution. This dynamic strategy ensures approximation quality while uses less particles on average. During the sampling, one should keep track of the number of bins that the approximated distribution occupies and estimate the number of samples needed accordingly.

## 2.4 Belief Packing

**Definition 1** ($\delta$-packing). Let $(X, \|\cdot\|)$ be a normed space and $Y \subseteq X$. $\{y_1, y_2, \ldots, y_n\} \subseteq Y$ is a $\delta$-packing of $Y$ if $\min_{i \neq j} \|y_i - y_j\| > \delta$. The $\delta$-packing number of $Y$, denoted $P^\delta(Y)$, is the maximum cardinality of any $\delta$-packing of $Y$.

*Remark.* If $X$ is totally bounded and $Y \subseteq X$, then we have $P^\delta(Y) < \infty$ for all $\delta > 0$.

A belief packing is a special case of $\delta$-packing, where the $X$ is the belief space, a probability simplex of $|\mathcal{S}| - 1$ dimensions, and the norm $\|\cdot\|$ is the $L1$ norm. Since a simplex is totally bounded, it follows that the packing number of a belief packing is finite.

## 2.5 Related Work

The covering number is the minimum number of balls needed to cover the entire space. It is closely related to the packing number and is extensively used to identify the complexity of POMDP problems [11, 16–18]. SARSOP [19] is an offline solver which builds an approximate cover of the beliefs space reachable by the optimal policy and performs point-based backup at beliefs in the cover. PGVI [17] improves SARSOP by maintaining belief packings and only expanding beliefs in the packing. FMP [20] applies the idea of PGVI to online planning and builds a $\delta$-cover of beliefs. Notwithstanding, all these methods only work in discrete state spaces and require enumerating all observations.

Online POMDP solvers mainly include POMCP [5], DESPOT [6], and their variants [7–9]. We will introduce them in terms of approximating beliefs and addressing large observation spaces.

On the belief approximation, DESPOT and POMCP sample from the joint distribution of states and observations and approximate a belief with samples having consistent observations. These methods are doomed to failure in large or continuous observation space because the same observation is unlikely to be produced twice. POMCPOW [7] weights particles according to the likelihood, which makes its belief updating close to SIR. Nonetheless, most of its beliefs are approximated by merely several particles causing the approximation to be inaccurate. PTF-DPW [7] updates beliefs with SIR particle filters. However, standard particle filters are excessively costly. The best performance of PTF-DPW is achieved when the number of particles used is only 20. DESPOT-$\alpha$ [9] and LABECOP [21] approximate beliefs with SIS, which is, as explained, ineffective compared to SIR.

On addressing observations, DESPOT and POMCP expand all sampled observations, which causes the observation branches to explode quickly. POMCPOW and PTF-DPW introduce the progressive

widening technique [22], which restricts the number of observation branches and gradually relaxes the restriction when a node is revisited multiple times. DESPOT-$\alpha$ instead fixes the maximum number of observation branches to be $C$. However, restricting the number of observation branches inevitably produces extra estimation variance, damaging the performance. LABECOP adopts the lazy belief extraction technique, which always extracts the belief from scratch with the newly sampled observation sequence. This method naturally comes with a higher computational cost.

# 3 Method

Online planning (Alg. 1) consists of two alternating stages, planning and execution. A look-ahead search is conducted during the planning stage to find the best action for the current belief. At the execution stage, the algorithm performs in the real environment the action found, gets a new observation, and updates the belief with a Bayes filter. Then a planning stage follows. AdaOPS follows the general procedure and makes changes in the planning part.

---
**Algorithm 1** General Procedure

**Input:** initial belief $\mathrm{bel}$
1: $b \leftarrow \mathrm{bel}$
2: **while** True **do**
3:     $a^* \leftarrow \text{PLANNING}(b)$
4:     Execute action $a^*$
5:     Receive observation $o$
6:     Update the belief $b$ with $a^*$ and $o$

---

## 3.1 Planning

At each timestep, AdaOPS gradually builds up a belief tree (as shown in Figure 1) consisting of alternating action and belief nodes, of which the root belief $\bar{b}_0$ is sampled from the current belief $b_0$ using KLD-Sampling. Each belief $\bar{b}$ on the tree is represented by a collection of weighted particles and is an approximation to a corresponding true belief $b$. Note that an approximated belief also corresponds to a belief node, and we will not distinguish between them in the following. Sibling beliefs on the tree share the particles and differ in the corresponding weights. Initially, the belief tree is comprised of the root node alone.

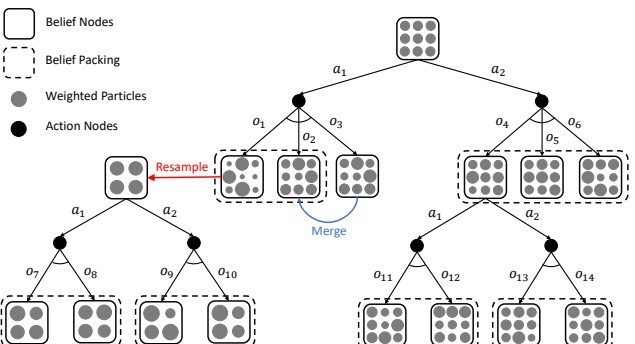

Figure 1: A belief tree built by AdaOPS. The sibling beliefs share the same particles and differ in the corresponding weights, forming a belief packing. The blue arrow indicates that similar nodes are merged. Beliefs with high-weight disparity will be resampled using KLD-Sampling as the red arrow shows.

A series of explorations will then be made, each traversing a path on the tree and expanding new nodes in the meantime. The complete pseudocode of AdaOPS is shown in Appendix A.

AdaOPS maintains an upper bound $u(\bar{b}, a)$ and a lower bound $l(\bar{b}, a)$ of the optimal value for every action node $a$ of belief $\bar{b}$. During the exploration, the action $a^*$ with the highest upper bound will be chosen (Line 10 in Alg. 2). This strategy ensures that an action branch gets explored only when its upper bound is higher than or equal to the optimal value. AdaOPS also maintains lower and upper bounds, $l(\bar{b})$ and $u(\bar{b})$, for each belief node $\bar{b}$. After choosing the action, it will choose (Line 11 in Alg. 2) the child belief with the highest probability-weighted excess uncertainty (EU) [6, 23], $\hat{p}(o \mid \bar{b}, a^*)\text{EU}(\tau(\bar{b}, a^*, o))$, where $\hat{p}(o \mid \bar{b}, a^*)$ is an estimation of the probability $\Pr(o \mid \bar{b}, a^*)$, and $\tau(\bar{b}, a^*, o)$ is the belief reached after executing $a^*$ and receiving $o$. The way to estimate $\Pr(o \mid \bar{b}, a^*)$ is stated in Section 3.2.1. The excess uncertainty is defined as

$$\text{EU}(\bar{b}) = u(\bar{b}) - l(\bar{b}) - \xi \cdot (u(\bar{b}_0) - l(\bar{b}_0))/\gamma^{\text{depth}(\bar{b})}, \tag{3}$$

where $\xi$ is a parameter for adjusting the desired gap between upper and lower bounds at the root and defaults to 0.95, and $\text{depth}(\bar{b})$ is the depth of belief $\bar{b}$ on the belief tree. The excess uncertainty measures how much uncertainty we need to eliminate at the belief node $\bar{b}$ if we want the gap at the root, $u(\bar{b}_0) - l(\bar{b}_0)$, to be reduced to $\xi \cdot (u(\bar{b}_0) - l(\bar{b}_0))$. The excess uncertainty has a desirable property: $\text{EU}(\bar{b}) \leq \sum_{o \in O_{\bar{b}, a^*}} \hat{p}(o \mid \bar{b}, a^*)\text{EU}(\tau(\bar{b}, a^*, o))$, where $O_{\bar{b}, a^*}$ is the set of expanded observations

**Algorithm 2** PLANNING($b_0$)

---

1: $\bar{b}_0 \leftarrow$ KLD-SAMPLING($b_0$)
2: **while** time permitting and $l(\bar{b}_0) < u(\bar{b}_0)$ **do**
3:      $\bar{b} \leftarrow \bar{b}_0$
4:      **while** depth($\bar{b}$) $< D$ **do**
5:          **if** $\bar{b}$ is a leaf node **then**
6:             EXPAND($\bar{b}$)
7:             BACKUP($\bar{b}$)
8:             **if** best action branch changes for some ancestors of $\bar{b}$ or EU($\bar{b}$) $\leq 0$ **then**
9:                 **break**
10:          $a^* \leftarrow \arg\max_{a \in \mathcal{A}} u(\bar{b}, a)$
11:          $o^* \leftarrow \arg\max_{o \in O_{\bar{b},a}} \hat{p}(o \mid \bar{b}, a)\text{EU}(\tau(\bar{b}, a^*, o))$
12:          $\bar{b} \leftarrow \tau(\bar{b}, a^*, o^*)$
13:      **if** depth($\bar{b}$) $\geq D$ **then**
14:          $u(\bar{b}) \leftarrow l(\bar{b})$
15: **return** $\arg\max_{a \in \mathcal{A}} l(\bar{b}_0, a)$

---

under the action branch $a^*$ of belief $b$. So, we can reduce the excess uncertainty on $b$ by reducing the excess uncertainty on its child beliefs. Note that minimizing the excess uncertainty on the root belief is equivalent to minimizing $u(\bar{b}_0) - l(\bar{b}_0)$.

Each time a leaf node is encountered (Line 6 in Alg. 2), all valid actions will be tried with new belief nodes expanded. For new belief nodes, the lower and upper bounds will be initialized with heuristics. This process of expanding leaf nodes is detailed in Section 3.2. After expanding a leaf node, a backup will be performed (Line 7 in Alg. 2) to update the bounds of ancestor nodes, which is realized by recursively applying the Bellman equation:

$$
\begin{aligned}
U(b) = \max_a U(b, a) &= \max_a \left\{ \int_{\mathcal{S}} b(s)R(s, a)\mathrm{d}s + \int_{\mathcal{O}} \Pr(o \mid b, a)U(\tau(b, a, o))\mathrm{d}o \right\} \\
&\approx U(\bar{b}) = \max_a \left\{ \sum_{s \in \mathcal{S}} \bar{b}(s)R(s, a) + \sum_{o \in O_{\bar{b},a}} \hat{p}(o \mid \bar{b}, a)U(\tau(\bar{b}, a, o)) \right\},
\end{aligned}
\tag{4}
$$

where $U(b)$ and $U(b, a)$ can be either upper or lower bound of the optimal value, $\bar{b}(s) = \sum_i \mathbb{I}(s_i = s)w_i$ is the probability of state $s$ estimated by the weighted particle collection of belief $\bar{b}$, and $\mathbb{I}(\cdot)$ is the indicator function. After backup, the upper bounds of ancestor nodes tend to decrease, approaching the optimal value. According to Equation (4), the upper bound of a belief node is only determined by its best action branch, i.e., the action branch with the highest upper bound. Therefore, if the best action branch changes for some ancestor nodes during the backup due to the decreasing of the upper bound, expanding the current branch will no longer decrease the upper bound of the root. In this case, AdaOPS will end the exploration and restart from the root node (Line 8 in Alg. 2). The exploration will also end if the excess uncertainty is less than or equal to 0 since we want to focus on branches with high uncertainty. Moreover, if a belief exceeds the maximum depth $D$ (Line 13 in Alg. 2), its upper bound will be set to its lower bound so as to prevent further searching.

With the process of explorations, the bounds on the root will progressively approach the optimal value. Nevertheless, the planning time is often not adequate for it to converge. When the allocated time is up, the highest lower-bound action at the root belief will be chosen for execution.

## 3.2 Expanding Leaf Nodes

During the expansion of leaf nodes, AdaOPS first checks if the resampling condition is met (Line 1 in Alg. 3). If it does, AdaOPS performs the KLD-Resampling [24], which means we apply the KLD-Sampling method to the resampling procedure. Then, for each action $a \in \mathcal{A}$, AdaOPS will propagate particles in belief $\bar{b}$ with a generative model (Line 4 in Alg. 3). This process is further discussed in Section 3.2.1. Particle propagation will produce a set of sampled observations and their corresponding beliefs, $B_{\bar{b},a}$, the average reward $R_{\bar{b},a}$ for performing action $a$ at belief $\bar{b}$, and an estimate $\hat{p}(\cdot \mid \bar{b}, a)$ for the probability $\Pr(o \mid \bar{b}, a)$. With the sample observations, AdaOPS generates

a belief packing (Line 5 in Alg. 3), $P_{\bar{b},a}$, as described in Section 3.2.2. Finally, all beliefs in the belief packing will be expanded to the belief tree, and their upper and lower bounds will be initialized with some value approximation techniques [25], which are outlined in Appendix B.

### 3.2.1 Adaptive Particle Filter

According to Equation (4), we need to approximate beliefs accurately so that the estimation for the first integral $\int_{\mathcal{S}} b(s)R(s,a)\mathrm{d}s \approx \sum_{s\in\mathcal{S}} \bar{b}(s)R(s,a)$ is accurate. However, in online planning, we cannot afford tons of particles for belief approximation and need a particle filter that works well with merely a few particles. That is why we introduce the adaptive resampling (Lines 1-2 in Alg. 3) and the KLD-Sampling method. When the sample size is small, the particle impoverishment issue will be prominent, which can be alleviated with adaptive resampling. Moreover, KLD-Sampling ensures the approximation quality while using fewer particles.

---

**Algorithm 3** EXPAND($\bar{b}$)

1: **if** $|\bar{b}|/\mathrm{ESS}(\bar{b}) > \mu$ **then**
2: $\quad \bar{b} \leftarrow$ KLD-SAMPLING($\bar{b}$)
3: **for** $a \in \mathcal{A}$ **do**
4: $\quad B_{\bar{b},a}, R_{\bar{b},a}, \hat{p}(\cdot \mid \bar{b}, a) \leftarrow$ PROPAGATE($\bar{b}, a$)
5: $\quad P_{\bar{b},a} \leftarrow$ GENERATEPACKING($B_{\bar{b},a}, \hat{p}(\cdot \mid \bar{b}, a)$)
6: $\quad$ Expand action node $a$
7: $\quad$ Expand beliefs in $P_{\bar{b},a}$ as child beliefs of $a$

---

**Algorithm 4** KLD-SAMPLING($\bar{b}$)

1: Count the number of bins with support for belief $\bar{b}$
2: Calculate the number $N$ satisfying Equation (2)
3: Sample $N$ particles in belief $\bar{b}$

---

AdaOPS performs the KLD-Resampling adaptively (Lines 1-2 in Alg. 3) when the particle number of the leaf belief $\bar{b}$ is greater than $\mu \, \mathrm{ESS}(\bar{b})$, where $\mu \geq 1$ adjusts the frequency of resampling and is set to 2 throughout the paper. According to Equation (1), a low effective sample size indicates a high level of weight disparity. It is also illustrated in Figure 1 that AdaOPS resamples when the weight disparity of a belief is high. In order to determine the particle number for resampling, the state space is partitioned into a grid (described in Appendix D.2, Figure 5). Then, we can count the number of bins occupied by belief $\bar{b}$ and calculate the particle number for resampling with Equation (2). The hyperparameter $\eta$ is set to be 0.05, and the $\zeta$ is tuned such that $m_{\min}$ is the minimal number satisfying Equation (2) when $k = 2$, i.e., $\zeta = \frac{1}{2m_{\min}} \left( 1 - \frac{2}{9} + \sqrt{\frac{2}{9}} z_{0.95} \right)^3$, where $m_{\min}$ is another hyperparameter representing the minimum number of particles needed for approximating the belief.

---

**Algorithm 5** PROPAGATE($\bar{b}, a$)

1: $O \leftarrow \varnothing, B \leftarrow \varnothing, R \leftarrow 0, \hat{p}(\cdot) \leftarrow 0$
2: Initialize $\tilde{b}$ as an empty weighted particle collection
3: **for** $(w, s) \in \bar{b}$ **do**
4: $\quad s', o, r \leftarrow G(s, a)$
5: $\quad \tilde{b} \leftarrow \tilde{b} \cup \{(w, s')\}, O \leftarrow O \cup \{o\}, R \leftarrow R + wr$
6: $\quad \hat{p}(o) \leftarrow \hat{p}(o) + w$
7: **for** $o \in O$ **do**
8: $\quad$ Initialize $\bar{b}'$ as an empty weighted particle collection
9: $\quad$ **for** $(w, s') \in \tilde{b}$ **do**
10: $\quad\quad \bar{b}' \leftarrow \bar{b}' \cup \{wZ(o \mid s', a)/\rho, s'\}$ $\qquad\qquad \triangleright \rho = \sum_{(w,s')\in\tilde{b}} wZ(o \mid s', a)$
11: $\quad B \leftarrow B \cup \{(o, \bar{b}')\}$
12: **return** $B, R, \hat{p}$

---

For each action $a \in \mathcal{A}$, AdaOPS propagates all particles $(w_i, s_i)$ of $\bar{b}$ through a generative model, i.e., $s'_i, o_i, r_i \leftarrow G(s_i, a)$ (Line 4 in Alg. 5) and calculates the average reward, $R = \sum_i w_i r_i$ (Line 5 in Alg. 5). We estimate the probability of generating an observation $o$, $\Pr(o \mid \bar{b}, a)$, by the sum of weights of particles generating it (Line 6 in Alg. 5), i.e., $\Pr(o \mid \bar{b}, a) \approx \hat{p}(o \mid \bar{b}, a) = \sum_i \mathbb{I}(o = o_i)w_i$.

The belief $\tau(\bar{b}, a, o)$ for sampled observation $o$ is approximated by the same collection of particles but assigns a different weight for each (Line 10 in Alg. 5). The weight for the $i$-th particle under

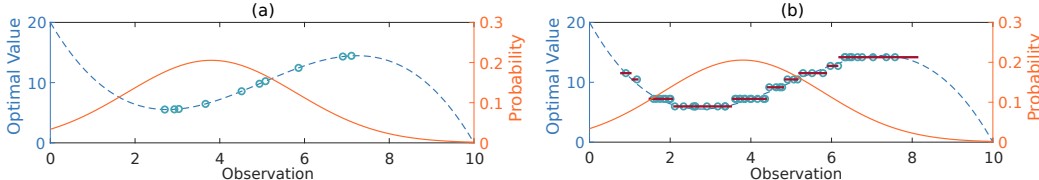

Figure 2: The solid orange line stands for an observation distribution $\Pr(\cdot \mid b, a)$ distributed between 0 to 10. The dashed blue line illustrates the optimal values of observations, $V^*(\tau(b, a, \cdot))$. (a) shows 10 samples from the observation distribution, forming an unbiased estimator for the first integral in Equation (4). Nonetheless, this estimator has a high variance since the sample size is too small. (b) shows 50 samples from the observation distribution. By building a belief packing, only 10 dissimilar beliefs having distinct optimal values are retained for evaluation. Each solid red line in the figure corresponds to a belief in the packing. The values of beliefs on the line are set to the value of the corresponding belief in the packing. By doing so, we obtain a low variance estimator with slight bias at the cost of 10 evaluations.

observation $o$ is $w'_i = w_i Z(o \mid s'_i, a)/\rho$, where $\rho = \sum_i w_i Z(o \mid s'_i, a)$ is a normalizing factor. The set of sampled observations and their corresponding beliefs is denoted as $B_{\bar{b},a}$.

### 3.2.2 Belief Packing

Equation (4) reveals that we can approximate the integral $\int_{\mathcal{O}} \Pr(o \mid b, a) U(\tau(b, a, o)) \mathrm{d}o$ with the value of some sampled observations. The more observations we sample, the lower the estimation variance is. However, each sampled observation corresponds to an observation branch. Suppose we expand all observations in $B_{\bar{b},a}$ as new branches, the induced branching factor is often beyond our computational capability. Nevertheless, restricting the number of observations renders high estimation variance. Inspired by [16–18], we introduce belief packing to fuse similar beliefs and only evaluate the distinct ones, balancing estimation bias and variance.

**Algorithm 6** GENERATEPACKING$(B, \hat{p})$

1: $P \leftarrow \varnothing$
2: **for** $(o, \bar{b}) \in B$ **do**
3:     covered $\leftarrow$ false
4:     **for** $(o', \bar{b}') \in P$ **do**
5:         **if** $\|\bar{b}' - \bar{b}\|_1 \leq \delta$ **then**
6:             $\hat{p}(o') \leftarrow \hat{p}(o') + \hat{p}(o)$
7:             $\hat{p}(o) \leftarrow 0$
8:             covered $\leftarrow$ true
9:             **break**
10:     **if** not covered **then**
11:         $P \leftarrow P \cup \{(o, \bar{b})\}$
12: **return** $P$

Since the sibling beliefs $\bar{b}$ and $\bar{b}'$ share the same particles, we can calculate their $L1$ distance with $\|\bar{b} - \bar{b}'\|_1 = \sum_{s \in \mathcal{S}} |\bar{b}(s) - \bar{b}(s')| = \sum_i |w_i - w'_i|$, where $w_i$ and $w'_i$ are the weights of the $i$-th particle for beliefs $\bar{b}$ and $\bar{b}'$, respectively. As will be shown in Lemma 1, when the belief approximation is accurate, and the $L1$ distance between two approximated beliefs is small, the true optimal value are also close. In other words, it is safe to fuse similar beliefs. The formal convergence analysis is given in Theorem 1.

In AdaOPS, a belief packing is maintained under each action branch $a$ of belief $\bar{b}$ as illustrated in Figure 1. We merge two observation branches when their beliefs are within a distance of $\delta$ (Line 5 in Alg. 6). Specifically, if the distance between $\bar{b}_1$ in the packing and $\bar{b}_2$ is within $\delta$, we set $\hat{p}(o_1 \mid \bar{b}, a) \leftarrow \hat{p}(o_1 \mid \bar{b}, a) + \hat{p}(o_2 \mid \bar{b}, a)$ and $\hat{p}(o_2 \mid \bar{b}, a) \leftarrow 0$ for their corresponding observations $o_1$ and $o_2$, which is equivalent to approximate the value of $\bar{b}_2$ with that of $\bar{b}_1$. If the belief $\bar{b}_2$ is not covered by any $\delta$-ball centered at beliefs in the packing, then we add it to the packing (Line 10 in Alg. 6).

As shown in Figure 2, by setting the value of a belief to that of the closest belief in the $\delta$-packing, we obtain a piece-wise constant approximation to $V^*(\tau(b, a, \cdot))$, which avoids evaluating $V^*(\cdot)$ at similar beliefs at the cost of introducing limited bias.

## 4 Theoretical Analysis

This section demonstrates some theoretical results and provides some insights on why and when AdaOPS works. The proof, along with a running time analysis, is detailed in Appendix C.

Assuming the reward function is Borel measurable and bounded, we denote $R_{\max} = \|R\|_\infty$. It is already proved in [11] that the value function of POMDPs satisfies the Lipschitz condition, i.e., $|V(b) - V(b')| \leq \frac{R_{\max}}{1-\delta}\delta$, if $\|b - b'\|_1 \leq \delta$. We present another theorem extending this conclusion to the case where we only have access to sets of weighted samples approximating the beliefs.

**Lemma 1.** By assigning the same set of $N$ samples different weights, we get $\bar{b}(s) = \sum_{i=1}^{N} w_i \mathbb{I}(s = s_i)$ and $\bar{b}'(s) = \sum_{i=1}^{N} w'_i \mathbb{I}(s = s_i)$ approximating beliefs $b$ and $b'$. There exist bounded functions $\alpha$ and $\alpha'$ such that $V^*(b) = \int \alpha(s)b(s)\,\mathrm{d}s$, and $V^*(b') = \int \alpha'(s)b'(s)\,\mathrm{d}s$. Supposing that $\|\bar{b} - \bar{b}'\|_1 \leq \delta$, $|V^*(b) - \sum_{i=1}^{N} w_i \alpha(s_i)| \leq \lambda$, and $|V^*(b') - \sum_{i=1}^{N} w'_i \alpha'(s_i)| \leq \lambda$, it follows that

$$|V^*(b) - V^*(b')| \leq 2\lambda + \frac{R_{\max}}{1-\gamma}\delta. \tag{5}$$

This result indicates that merging similar beliefs works well when the approximated beliefs are indeed good approximations to the original beliefs.

In the following, we will present the convergence analysis of AdaOPS. In other words, we will bound the difference between the real optimal value $V_0^*(b_0)$ of the current belief $b_0$ and the estimated optimal value $\hat{V}_0^*(\bar{b}_0)$ of the root belief $\bar{b}_0$. As the resampling will cause the particles to be interrelated and significantly increase the difficulty of analysis, we follow the proof in [26] and disables the resampling for now. The effectiveness of the adaptive particle filter is empirically demonstrated in Section 5.2. Our convergence analysis is built upon the following assumptions:

1. The ESS threshold $\mu$ for adaptive resampling is set to $\infty$.
2. The reward function is Borel measurable and bounded.
3. The upper bound of AdaOPS is initialized to $+\infty$. Its lower bound is initialized with any function bounded by $\frac{R_{\max}}{1-\gamma}$.
4. The state and observation spaces are continuous, and the action space is finite.
5. For all $d \in \{0, 1, \ldots, D-1\}$ and all observation sequences $\{o_n\}$, the essential supremum of the importance weight $\operatorname{ess\,sup}_{x \sim \mathcal{Q}^d} w_{\mathcal{P}_{\{o_n\}}^d/\mathcal{Q}^d}(x)$ is upper bounded by $d_{\max}$, where $w_{\mathcal{P}/\mathcal{Q}}(x) = \frac{\mathcal{P}(x)}{\mathcal{Q}(x)}$.

**Theorem 1.** Suppose that all assumptions listed above hold. Let $N = |\bar{b}_0| \geq m_{\min}$. For $\epsilon > 0$,

$$|V_0^*(b_0) - \hat{V}_0^*(\bar{b}_0)| \leq \epsilon \tag{6}$$

holds with a probability of at least

$$1 - 15|\mathcal{A}|N(|\mathcal{A}|\min(N, P_{\max}^\delta))^D \exp\left(-N \cdot t_{\max}^2\right), \tag{7}$$

where $t_{\max} = \frac{(1-\gamma)\lambda}{R_{\max}d_{\max}} - \frac{1}{\sqrt{N}}$, $\delta = \frac{(1-\gamma)^2\epsilon}{3\gamma R_{\max}}$, $\lambda = \frac{(1-\gamma)\epsilon}{3(1+5\gamma/2)}$, $D = \log_\gamma \frac{(1-\gamma)\epsilon}{6R_{\max}}$, $P_{\max}^\delta = \sup_{b,a} P^\delta(Y_{b,a})$, and $Y_{b,a}$ is the set of reachable beliefs after executing action $a$ at belief $b$.

The sample size $N$ in Theorem 1 is at an order of $\tilde{O}\left(\frac{R_{\max}^2 d_{\max}^2}{(1-\gamma)^4\epsilon^2}\right)$, where the tilde means to omit the logarithmic part. Since the term $d_{\max}$ can increase exponentially w.r.t. the depth $D$, the sample size could scale poorly, highlighting the importance of sample-efficient belief approximation.

The benefit of belief packing is not evident from the convergence result. In fact, it affects the performance by controlling the tree size. AdaOPS builds a tree of size $O((|\mathcal{A}|\min(P_{\max}^\delta, N_{\max}))^D)$, where $N_{\max}$ is the maximum sample size given by KLD-Sampling. For discrete state spaces, the maximum packing number $P_{\max}^\delta$ is always finite and will not increase w.r.t. depth $D$. Although this may not hold in every continuous state problem, belief packing still shows great potential empirically.

## 5 Experiments

This section evaluates our method on four domains based on the POMDPs.jl framework [27] and conducts an ablation study showing the contribution of each component. Besides, we compare the adaptive particle filter with other particle filter variants demonstrating that the adaptive particle filter can approximate the belief better.

## 5.1 Performance Comparison

In what follows, we compare AdaOPS with DESPOT [6] and POMCPOW [7] on four domains, Laser Tag [6], Rock Sample (RS) [23], Roomba [28], and Light Dark [7]. These domains are carefully chosen so that they can cover various types of POMDPs. The details of these domains are shown in Appendix D.1. In order to understand the contribution from adaptive resampling (AR), KLD-Sampling (KLD-S), and belief packing (BP), respectively, we remove one component at a time and tune the hyperparameters to achieve the best performance for others. Specifically, the removal of adaptive resampling means we resample at each timestep as a standard particle filter does.

### 5.1.1 Experiment settings

In all these tasks, solvers are allocated $1s$ for planning per step. Since the obstacle positions in Laser Tag and the rock positions in Rock Sample greatly influence the optimal value, tests on these domains are conducted with one hundred randomly generated maps so that the result is trustworthy. Hyperparameters for each algorithm are tuned via grid search, and another independent experiment is conducted to test their performance. DESPOT and AdaOPS initialize upper and lower bounds with heuristics, and POMCPOW evaluates nodes it encountered for the first time with a value estimator. All experiments were conducted on a computer with Intel(R) Core(TM) i7-10750H, 6vCPUs running at 2.6GHz, and 32G main memory. Please refer to Appendix D.2 for the ranges of grid search and the hyperparameters and heuristics selected for each algorithm. Besides, a time sensitivity analysis is presented in Appendix D.3.

### 5.1.2 Results

In Table 1, the average discounted return is shown along with the corresponding standard error of mean (SEM) in the form of $\mathrm{Return} \pm \mathrm{SEM}$. In all these domains, AdaOPS outperforms other solvers with statistical significance ($p < 0.0001$). A hyperparameter sensitivity analysis in Light Dark is presented in Appendix D.4.

Table 1: Performance Comparison

|  | Laser Tag | RS(15,15) | Bumper Roomba | Lidar Roomba | Light Dark |
|---|---|---|---|---|---|
| $\lvert\mathcal{S}\rvert$ | $4,830$ | $7,372,800$ | $\infty$ | $\infty$ | $\infty$ |
| $\lvert\mathcal{A}\rvert$ | $5$ | $20$ | $9$ | $6$ | $3$ |
| $\lvert\mathcal{O}\rvert$ | $\sim 1.5 \times 10^6$ | $3$ | $2$ | $\infty$ | $\infty$ |
| DESPOT | $-10.71 \pm 0.22$ | $15.67 \pm 0.20$ | $-2.18 \pm 0.11$ | $-0.84 \pm 0.06$ | $2.50 \pm 0.10$ |
| POMCPOW | $-10.15 \pm 0.17$ | $10.40 \pm 0.18$ | $-2.28 \pm 0.09$ | $-0.50 \pm 0.07$ | $3.23 \pm 0.11$ |
| AdaOPS | $\mathbf{-8.31 \pm 0.18}$ | $\mathbf{17.16 \pm 0.21}$ | $\mathbf{-1.48 \pm 0.11}$ | $\mathbf{-0.09 \pm 0.07}$ | $\mathbf{3.79 \pm 0.07}$ |
| AdaOPS(AR) | $-8.79 \pm 0.17$ | $16.73 \pm 0.21$ | $-1.63 \pm 0.11$ | $-0.21 \pm 0.06$ | $3.62 \pm 0.07$ |
| AdaOPS(KLD-S) | $-8.79 \pm 0.18$ | $\backslash$ | $-1.71 \pm 0.11$ | $-0.20 \pm 0.07$ | $3.72 \pm 0.07$ |
| AdaOPS(BP) | $-9.84 \pm 0.18$ | $17.10 \pm 0.21$ | $\backslash$ | $-0.51 \pm 0.07$ | $1.50 \pm 0.20$ |

Note: RS(n,m) stands for the Rock Sample with $n \times n$ map and $m$ rocks. $\infty$ means continuous state (or observation) space.

AdaOPS(XX) denotes AdaOPS without component XX. AR = Adaptive Resampling. KLD-S = KLD-Sampling. BP = Belief Packing.

**Laser Tag**  In Laser Tag, an agent tries to tag an escaping target. Initially, the agent knows nothing about its own position and the target's position, and the belief is spread over the entire grid. It can then access the map and infer the positions from the sensor information. After several steps, the belief will shrink into a small region. Since the dispersion of beliefs varies considerably, the KLD-Sampling is advantageous. The adaptive resampling avoids sample impoverishment and contributes equally with KLD-Sampling. The belief packing contributes most by allowing AdaOPS to handle massive observation space effectively.

**Rock Sample**  In Rock Sample, a Mars rover intends to sample as many good rocks as possible while only inferring the goodness of rocks from its noisy sensor measurements. It maintains a belief of the goodness for each rock, which is always comprised of two bins, good and bad. Thus, the particle number given by Equation (2) is always $m_{\min}$, i.e., KLD-Sampling makes no difference in

the Rock Sample problem. Besides, belief packing is also unproductive since there are only two observations. The improvements of AdaOPS mainly come from the fact that AdaOPS maintains an accurate belief approximation in the depth of the tree using particle filter techniques. Contrarily, DESPOT and POMCPOW tend to be overoptimistic on future uncertainty.

**Roomba**  Roomba is a robotic vacuum cleaner that attempts to locate itself in a familiar room and reach the target region. It can equip either a Bumper sensor that senses the collision or a Lidar sensor that measures the distance to the obstacle in the front. As a state is either collided or not, the beliefs corresponding to different observations in Bumper Roomba always have an $L1$ distance 2, which means there are no similar beliefs, and belief packing is of little avail in this case. However, in Lidar Roomba, belief packing handles continuous observation gracefully, making a significant contribution. The KLD-Sampling fails to contribute much as expected in the Roomba domain. A possible reason is that according to Equation (2), high state uncertainty comes with a large number of particles and, consequently, observations, making online planning extremely hard given limited planning time. The above analysis is corroborated by an additional result presented in Appendix D.5.

**Light Dark**  Light Dark is an information-gathering task. With the aim of reaching the target area, the agent needs to find its way to the light region, gathering intelligence. This requires the agent to be fully aware of the state uncertainty while factoring in all possible observations. The belief packing empowers AdaOPS to consider only distinct observations. On the KLD-Sampling, the problem in Roomba also occurs in Light Dark, resulting in little improvement.

## 5.2  Benefits of Adaptive Particle Filters

We present a comparison between different particle filters on the Laser Tag domain, including SIS, SIR, and Adaptive particle filter. Each particle filter is given the same sequence of action-observation history for belief updating. The approximation error (measured by total variation distance) is checked after 10 steps. As illustrated in Figure 3, the adaptive particle filter gets the lowest approximation error and the lowest variance using the same amount of particles. This result supplements the convergence analysis in Section 4 since a better approximation of beliefs will not undermine the convergence.

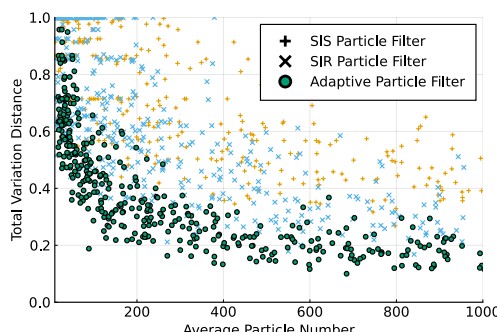

Figure 3: The belief approximated via the adaptive particle filter is the closest to the real posterior, using the same amount of particles.

## 6  Discussion

This paper presents an online planning algorithm, AdaOPS, featuring adaptive particle filter and belief packing, which is able to solve most POMDP problems optimally, given enough planning time. The success of the adaptive particle filter points out that in previous methods [6, 7], the inaccurate belief updating is a detriment to the performance, outweighing the potential benefit of efficiency. Furthermore, it is demonstrated that forming a belief packing with approximated beliefs is an effective tool for handling large observation spaces. We believe it could find itself extensive use in POMDP planning.

The size of the tree AdaOPS built could go unmanageable when $|\mathcal{A}|$ and $P_{\max}^{\delta}$ are large. Though it is caused by the complexity of the problem itself [11], we can abate this problem by implementing a parallel version following [8]. Furthermore, it is possible to improve the search efficiency with sawtooth approximation [25] and alpha vectors as described in [9]. We will also explore the possibility of handling large-scale or continuous action spaces.

## Acknowledgements

We thank Haoran Xiang, Dongyu Guo, Weijian Liao, and Feng Xu for helpful discussions and support in this project. This work was in part supported by the NSFC (61876119), the Fundamental Research Funds for the Central Universities (022114380010), Huawei Noah's Ark Lab (HF2019105005), and Alibaba Group through Alibaba Research Fellowship Program.

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
