# Appendix

## A  Pseudocode for AdaOPS

---

**Algorithm 7** AdaOPS

---

**Input:**
 bel: Initial belief.
 $\xi$: The parameter for adjusting the desired gap between upper and lower bounds at the root.
 $\delta$: The $\delta$-packing of beliefs will be generated.
 $m_{\min}$: The minimum number of particles for approximating beliefs.
 $D$: The maximum depth of the belief tree.
 $t$: The planning time allowance per step.

1: $b \leftarrow$ bel
2: **while** True **do**
3:     $a^* \leftarrow$ PLANNING($b$)
4:     Execute action $a^*$
5:     Receive observation $o$
6:     Update the belief $b$ with $a^*$ and $o$

7: **function** PLANNING($b_0$)
8:     $\bar{b}_0 \leftarrow$ KLD-SAMPLING($b_0$)
9:     **while** time permitting, $l(\bar{b}_0) < u(\bar{b}_0)$ **do**
10:         $\bar{b} \leftarrow \bar{b}_0$
11:         **while** depth($\bar{b}$) $< D$ **do**
12:             **if** $\bar{b}$ is a leaf node **then**
13:                 EXPAND($\bar{b}$)
14:                 BACKUP($\bar{b}$)
15:                 **if** current branch is suboptimal or EU($\bar{b}$) $\leq 0$ **then**
16:                     **break**
17:                 **end if**
18:             $a^* \leftarrow \arg\max_{a \in \mathcal{A}} u(\bar{b}, a)$
19:             $o^* \leftarrow \arg\max_{o \in O_{\bar{b},a}} \hat{p}(o \mid \bar{b}, a)\text{EU}(\tau(\bar{b}, a^*, o))$
20:             $\bar{b} \leftarrow \tau(\bar{b}, a^*, o^*)$
21:         **if** depth($\bar{b}$) $\geq D$ **then**
22:             $u(\bar{b}) \leftarrow l(\bar{b})$
23:     **return** $\arg\max_{a \in \mathcal{A}} l(\bar{b}_0, a)$

24: **function** EXPAND($\bar{b}$)
25:     **if** $\lfloor\bar{b}\rfloor/\text{ESS}(\bar{b}) > \mu$ **then**
26:         $\bar{b} \leftarrow$ KLD-SAMPLING($\bar{b}$)
27:     **for** $a \in \mathcal{A}$ **do**
28:         $B_{\bar{b},a}, R_{\bar{b},a}, \hat{p}(\cdot) \leftarrow$ PROPAGATE($\bar{b}, a$)
29:         $P_{\bar{b},a} \leftarrow$ GENERATEPACKING($B_{\bar{b},a}, \hat{p}(\cdot)$)
30:         Expand action node $a$

31:         Expand beliefs in $P_{\bar{b},a}$ as child beliefs of $a$

32: **function** BACKUP($x$)
33:     **for all** $\bar{b}, a^*$ from $x$ to the root **do**
34:         Update $u(\bar{b}, a^*), l(\bar{b}, a^*), u(\bar{b}), l(\bar{b})$ by Equation (4)

35: **function** KLD-SAMPLING($\bar{b}$)
36:     Count the number of bins with support for belief $\bar{b}$
37:     Calculate $N$ using Equation (2)
38:     Sample $N$ particles in belief $\bar{b}$

39: **function** PROPAGATE($\bar{b}, a$)
40:     $O \leftarrow \varnothing, B \leftarrow \varnothing, R \leftarrow 0, \hat{p}(\cdot) \leftarrow 0$
41:     Initialize $\tilde{b}$ as an empty weighted particle collection
42:     **for** $(w, s) \in \bar{b}$ **do**
43:         $s', o, r \leftarrow G(s, a)$
44:         $\tilde{b} \leftarrow \tilde{b} \cup \{(w, s')\}, O \leftarrow O \cup \{o\}$
45:         $R \leftarrow R + wr$
46:         $\hat{p}(o) \leftarrow \hat{p}(o) + w$
47:     **for** $o \in O$ **do**
48:         Initialize $\bar{b}'$ as an empty weighted particle collection
49:         $\rho = \sum_{(w,s') \in \tilde{b}} wZ(o \mid s', a)$
50:         **for** $(w, s') \in \tilde{b}$ **do**
51:             $\bar{b}' \leftarrow \bar{b}' \cup \{wZ(o \mid s', a)/\rho, s'\}$
52:         $B \leftarrow B \cup \{(o, \bar{b}')\}$
53:     **return** $B, R, \hat{p}$

54: **function** GENERATEPACKING($B, \hat{p}$)
55:     $P \leftarrow \varnothing$
56:     **for** $(o, \bar{b}) \in B$ **do**
57:         covered $\leftarrow$ false
58:         **for** $(o', \bar{b}') \in P$ **do**
59:             **if** $||\bar{b}' - \bar{b}||_1 \leq \delta$ **then**
60:                 $\hat{p}(o') \leftarrow \hat{p}(o') + \hat{p}(o)$
61:                 $\hat{p}(o) \leftarrow 0$
62:                 covered $\leftarrow$ true
63:                 **break**
64:         **if** not covered **then**
65:             $P \leftarrow P \cup \{(o, \bar{b})\}$
66:     **return** $P$

---

# B  Bounds Initialization

Since AdaOPS maintains an approximation for all belief nodes it expanded, it can utilize all kinds of value approximation techniques to initialize its upper and lower bounds. This paper takes the simplest choice, fixed strategy approximation for lower bounds and the QMDP [29] or MDP approximation for upper bounds.

**Lower Bound Initialization**   The blind policy is a policy that takes the same action regardless of the observations it receives. A blind policy approximation can be represented as $|\mathcal{A}|$ alpha vectors,

$$\alpha_a(s) = R(s, a) + \gamma \sum_{s' \in \mathcal{S}} T(s' \mid s, a)\alpha_a(s'),$$

where $\alpha_a$ is the alpha vector corresponds to action $a$. With these alpha vectors, $l(\bar{b}) = \arg\max_{a \in \mathcal{A}} \sum_{s \in \mathcal{S}} \bar{b}(s)\alpha_a(s)$ constitutes a value lower bound for belief $\bar{b}$. Fixed action policy simplifies the blind policy comprising only one alpha vector $\alpha_a$. It provides a lower bound, $l(\bar{b}) = \sum_{s \in \mathcal{S}} \bar{b}(s)\alpha_a(s)$. Similarly, a random policy can also be represented as an alpha vector,

$$\alpha_{\text{rand}}(s) = \frac{1}{|\mathcal{A}|} \sum_{a \in \mathcal{A}} \left( R(s, a) + \gamma \sum_{s' \in \mathcal{S}} T(s' \mid s, a)\alpha_{\text{rand}}(s') \right).$$

**Upper Bound Initialization**   The MDP approximation assumes full observability and can be described by an alpha vector,

$$\alpha^{\text{MDP}}(s) = \max_{a \in \mathcal{A}} \left( R(s, a) + \gamma \sum_{s \in \mathcal{S}} T(s' \mid s, a)\alpha^{\text{MDP}}(s') \right).$$

The QMDP [29] approximation assumes the full observability after the first step and is represented by a set of $|\mathcal{A}|$ alpha vectors,

$$\alpha^a(s) = R(s, a) + \gamma \sum_{s \in \mathcal{S}} T(s' \mid s, a) \max_{a' \in \mathcal{A}} \alpha^{a'}(s'),$$

where $\alpha^a$ is the QMDP alpha vector corresponding to action $a$.

# C  Theoretical Analysis

## C.1  Termination

The particle number given by KLD-Sampling is bounded because the number of multidimensional bins is finite. According to Alg. 2, in each exploration, at least one leaf node will be expanded. Moreover, the overall size of the belief tree is $O((|\mathcal{A}| \min(P_{\max}^{\delta}, N_{\max}))^D)$, where $N_{\max}$ is the maximum sample size given by KLD-Sampling, $P_{\max}^{\delta} = \sup_{b,a} P^{\delta}(Y_{b,a})$, and $Y_{b,a}$ is the set of reachable beliefs after executing action $a$ at belief $b$. The tree size is limited since $N_{\max}$ is finite. Thus, we have the conclusion that AdaOPS is guaranteed to terminate.

When the algorithm terminates, as indicated by Equation (3), it is guaranteed that the gap at the root is 0. Since the upper and lower bounds are equal, we denote the value at a belief node $\bar{b}$ as $\tilde{V}^*(\bar{b})$.

## C.2  Convergence

**Lemma 1.** By assigning the same set of $N$ samples different weights, we get $\bar{b}(s) = \sum_{i=1}^{N} w_i \mathbb{I}(s = s_i)$ and $\bar{b}'(s) = \sum_{i=1}^{N} w_i' \mathbb{I}(s = s_i)$ approximating beliefs $b$ and $b'$. The weights are normalized, i.e., $\sum_{i=1}^{N} w_i = \sum_{i=1}^{N} w_i' = 1$. There exist bounded functions $\alpha$ and $\alpha'$ such that $V^*(b) = \int \alpha(s)b(s)\,\mathrm{d}s$, and $V^*(b') = \int \alpha'(s)b'(s)\,\mathrm{d}s$. Supposing that $\|\bar{b} - \bar{b}'\|_1 = \sum_{i=1}^{N} |w_i - w_i'| \leq \delta$, $|V^*(b) - \sum_{i=1}^{N} w_i \alpha(s_i)| \leq \lambda$, and $|V^*(b') - \sum_{i=1}^{N} w_i' \alpha'(s_i)| \leq \lambda$, it follows that

$$|V^*(b) - V^*(b')| \leq 2\lambda + \frac{R_{\max}}{1 - \gamma}\delta. \tag{8}$$

*Proof.* First, we will demonstrate that the value of any belief can be formulated as an integral. The optimal value of belief $b$ can be written as

$$V^*(b) = \max_a Q^*(b,a) = \max_a \int (R(s,a) + \gamma \mathbf{V}^*(s,b,a)) b(s) \, \mathrm{d}s$$

$$= \int \alpha(s) b(s) \, \mathrm{d}s, \tag{9}$$

where $\alpha(s) = R(s,a^*) + \gamma \mathbf{V}^*(s,b,a^*)$, $a^*$ is the optimal action, and

$$\mathbf{V}^*(s,b,a) = \int_{\mathcal{S}} \int_{\mathcal{O}} V^*(\tau(b,a,o)) Z(o|a,s') T(s'|s,a) \, \mathrm{d}s' \, \mathrm{d}o. \tag{10}$$

It is worth noting that $\alpha$ is bounded by $\frac{R_{\max}}{1-\gamma}$.

Hence, we can write $V^*(b)$ and $V^*(b')$ as $\int \alpha(s) b(s) \, \mathrm{d}s$ and $\int \alpha'(s) b(s) \, \mathrm{d}s$, respectively. The value difference $|V^*(b) - V^*(b')|$ can then be separated into three terms as follows,

$$|V^*(b) - V^*(b')| \leq \left| V^*(b) - \sum_{i=1}^N w_i \alpha(s_i) \right| + \left| \sum_{i=1}^N w_i \alpha(s_i) - \sum_{i=1}^N w_i' \alpha'(s_i) \right|$$

$$+ \left| \sum_{i=1}^N w_i' \alpha'(s_i) - V^*(b') \right|. \tag{11}$$

We can bound the first and third terms, respectively, by $\lambda$ in light of the assumptions. The second term is bounded following the proof presented in [11]. Let $w_i^c = \beta w_i + (1-\beta) w_i'$ such that $\sum_{i=1}^N w_i^c \alpha(s_i) = \sum_{i=1}^N w_i^c \alpha'(s_i)$, where $\beta \in [0,1]$. It can be shown that

$$\left| \sum_{i=1}^N w_i \alpha(s_i) - \sum_{i=1}^N w_i' \alpha'(s_i) \right|$$

$$= \left| \sum_{i=1}^N w_i \alpha(s_i) - \sum_{i=1}^N w_i^c \alpha(s_i) + \sum_{i=1}^N w_i^c \alpha'(s_i) - \sum_{i=1}^N w_i' \alpha'(s_i) \right|$$

$$= \left| \sum_{i=1}^N (1-\beta) \alpha(s_i)(w_i - w_i') + \sum_{i=1}^N \beta \alpha'(s_i)(w_i - w_i') \right| \tag{12}$$

$$= \left| \sum_{i=1}^N ((1-\beta)\alpha(s_i) + \beta\alpha'(s_i))(w_i - w_i') \right|$$

$$\leq \frac{R_{\max}}{1-\gamma} \sum_{i=1}^N |w_i - w_i'| \leq \frac{R_{\max}}{1-\gamma} \delta.$$

The proof is completed. $\qquad\qquad\square$

**Lemma 2.** [26] Let $\mathcal{P}$ and $\mathcal{Q}$ be two probability measures on the measurable space $(\mathcal{X}, \mathcal{F})$ with $\mathcal{P} \ll \mathcal{Q}$ and $d = \mathrm{ess\,sup}_{x \sim \mathcal{Q}} w_{\mathcal{P}/\mathcal{Q}}(x) < +\infty$. Let $x_1, \ldots, x_N$ be i.i.d.r.v. sampled from $\mathcal{Q}$, and $f : \mathcal{X} \to \mathbb{R}$ be a bounded Borel function ($\|f\|_\infty < +\infty$). Then, for any $\lambda > 0$ and $N$ large enough such that $\lambda > \|f\|_\infty d/\sqrt{N}$, the following bound holds with probability at least $1 - 3\exp(-N \cdot t^2(\lambda, N))$:

$$|\mathbb{E}_{x \sim \mathcal{P}}[f(x)] - \tilde{\mu}_{\mathcal{P}/\mathcal{Q}}| \leq \lambda, \tag{13}$$

where $t(\lambda, N)$ is defined as:

$$t(\lambda, N) = \frac{\lambda}{\|f\|_\infty d} - \frac{1}{\sqrt{N}}. \tag{14}$$

This lemma is a concentration inequality of self-normalized importance sampling estimator. It will enable us to establish the concentration of the value estimate in AdaOPS.

**Theorem 1.** Suppose that all assumptions listed below hold:

1. The ESS threshold $\mu$ for adaptive resampling is set to $\infty$.

2. The reward function is Borel measurable and bounded, and $R_{\max} = \|R\|_\infty$.

3. The upper bound of AdaOPS is initialized to $+\infty$. Its lower bound is initialized with any function bounded by $\frac{R_{\max}}{1-\gamma}$.

4. The state and observation spaces are continuous, and the action space is finite.

5. For all $d \in \{0, 1, \ldots, D-1\}$ and all observation sequences $\{o_n\}$, the essential supremum of the importance weight $\operatorname{ess\,sup}_{x \sim \mathcal{Q}^d} w_{\mathcal{P}^d_{\{o_n\}}/\mathcal{Q}^d}(x)$ is upper bounded by $d_{\max}$, where $w_{\mathcal{P}/\mathcal{Q}}(x) = \frac{\mathcal{P}(x)}{\mathcal{Q}(x)}$.

Suppose that all assumptions listed above hold. Let $N = |\bar{b}_0| \geq m_{\min}$. For $\epsilon > 0$,

$$|V_0^*(b_0) - \hat{V}_0^*(\bar{b}_0)| \leq \epsilon \tag{15}$$

holds with a probability of at least

$$1 - 15|\mathcal{A}|N(|\mathcal{A}|\min(N, P^\delta_{\max}))^D \exp\left(-N \cdot t_{\max}^2\right), \tag{16}$$

where $t_{\max} = \frac{(1-\gamma)\lambda}{R_{\max}d_{\max}} - \frac{1}{\sqrt{N}}$, $\delta = \frac{(1-\gamma)^2\epsilon}{3\gamma R_{\max}}$, $\lambda = \frac{(1-\gamma)\epsilon}{3(1+5\gamma/2)}$, $D = \log_\gamma \frac{(1-\gamma)\epsilon}{6R_{\max}}$, $P^\delta_{\max} = \sup_{b,a} P^\delta(Y_{b,a})$, and $Y_{b,a}$ is the set of reachable beliefs after executing action $a$ at belief $b$.

*Proof.* The assumption on the ESS threshold $\mu$ ensures that the resampling is turned off since the effective sample size is less than or equal to the actual sample size. After the initial resampling (Line 1 in Alg. 2), the particle number for belief approximation will be fixed to a constant $N \geq m_{\min}$. The restriction on the bounds initialization makes sure that the estimated lower bound will not surpass the estimated upper bound until all beliefs up to depth $D$ are expanded. Without this assumption, we can still derive a similar conclusion by handling the error introduced when an estimated lower bound wrongly surpasses the estimated upper bound.

During the planning, a set of states are sampled from the current belief $b_0$. These samples are then propagated according to the transition function. With the action sequence fixed, this process can be considered as sampling from a proposal distribution $\mathcal{Q}^d(\{s_n\}) = T_{1:d}b_0$, where $T_{1:d} = \prod_{n=1}^d T(s_n|s_{n-1}, a_n)$ is the transition density of state sequence $\{s_n\}$ given the action sequence. However, the interested target distribution is the posterior distribution for a given observation sequence $\{o_n\}$,

$$\mathcal{P}^d_{\{o_n\}}(\{s_n\}) = \frac{Z^{\{o_n\}}_{1:d}T_{1:d}b_0}{\int_{\mathcal{S}^{d+1}} Z^{\{o_n\}}_{1:d}T_{1:d}b_0 \, \mathrm{d}s_{0:d}}, \tag{17}$$

where $Z^{\{o_n\}}_{1:d} = \prod_{n=1}^d Z(o_n|a_n, s_n)$ is the conditional density of the observation sequence given the state and action sequence. The belief at depth $d$ can be obtained by marginalizing out $s_{0:d-1}$ from distribution $\mathcal{P}^d_{\{o_i\}}$. The approximation of the target distribution can be obtained by reweighting the proposal samples with the following self-normalized importance sampling weights:

$$w_{\mathcal{P}^d/\mathcal{Q}^d_{\{o_n\}}}(\{s_n\}) = \frac{Z^{\{o_n\}}_{1:d}}{\int_{\mathcal{S}^{d+1}} Z^{\{o_n\}}_{1:d}T_{1:d}b_0 \, \mathrm{d}s_{0:d}} \propto Z^{\{o_n\}}_{1:d}. \tag{18}$$

For a node at depth $D$, its value estimation error is bounded by $2\frac{R_{\max}}{1-\gamma}$, i.e.,

$$|V_D^*(b_D) - \hat{V}_D^*(\bar{b}_D)| \leq \frac{2R_{\max}}{1-\gamma} = \epsilon_D \tag{19}$$

holds for any belief $b_D$ at depth $D$. Here, $\bar{b}_d$ represents the particle approximation of belief $b_d$.

With $d = D$ as the base case, we bound the estimation error via backward induction. For $0 \leq d \leq D - 1$, we have

$$|Q_d^*(b_d, a) - \hat{Q}_d^*(\bar{b}_d, a)|$$

$$\leq \underbrace{\left| \mathbb{E}[R(s_d, a)|b_d] - \sum_{i=1}^{N} w_{d,i} r_{d,i} \right|}_{(A)} + \gamma \underbrace{\left| \mathbb{E}[V_{d+1}^*(\tau(b, a, o))|b_d] - \sum_{i=1}^{N} w_{d,i} \tilde{V}_{d+1}^*(\bar{b}_d, a, o_i) \right|}_{(B)}, \quad (20)$$

where $w_{d,i}$ is the importance weight of the $i$-th sample at depth $d$, and

$$\tilde{V}_{d+1}(\bar{b}_d, a, o_i) = \begin{cases} \hat{V}_{d+1}(\tau(\bar{b}_d, a, o_i)) & \text{if } \tau(\bar{b}_d, a, o_i) \in P_{\bar{b}_d, a}, \\ \hat{V}_{d+1}(\text{NN}(\tau(\bar{b}_d, a, o_i), P_{\bar{b}_d, a})) & \text{otherwise.} \end{cases} \quad (21)$$

Here, $\text{NN}(\bar{b}, P)$ is the nearest neighbor of $\bar{b}$ in the set $P$ measured by the $L1$ distance, and $P_{\bar{b}_d, a}$ is the belief packing built under the action branch $a$ of belief $\bar{b}_d$.

(A) is bounded by $(1 - \gamma)\lambda$ with a probability of at least $1 - \eta$ according to Lemma 2, where $\eta = 3 \exp(-N \cdot t_{\max}^2)$, and

$$t_{\max} = \frac{(1 - \gamma)\lambda}{R_{\max} d_{\max}} - \frac{1}{\sqrt{N}}. \quad (22)$$

For (B), we first separate it into three terms,

$$\left| \mathbb{E}[V_{d+1}^*(\tau(b_d, a, o))|b_d] - \sum_{i=1}^{N} w_{d,i} \tilde{V}_{d+1}^*(\bar{b}_d, a, o_i) \right|$$

$$\leq \left| \mathbb{E}[V_{d+1}^*(\tau(b_d, a, o))|b_d] - \sum_{i=1}^{N} w_{d,i} \mathbf{V}_{d+1}^*(s_{d,i}, b_d, a) \right|$$

$$+ \left| \sum_{i=1}^{N} w_{d,i} \left( \mathbf{V}_{d+1}^*(s_{d,i}, b_d, a) - V_{d+1}^*(\tau(b_d, a, o_i)) \right) \right| \quad (23)$$

$$+ \left| \sum_{i=1}^{N} w_{d,i} \left( V_{d+1}^*(\tau(b_d, a, o_i)) - \tilde{V}_{d+1}^*(\bar{b}_d, a, o_i) \right) \right|,$$

where

$$\mathbf{V}_{d+1}^*(s_{d,i}, b_d, a) = \int_{\mathcal{S}} \int_{\mathcal{O}} V_{d+1}^*(\tau(b_d, a, o)) Z(o|a, s_{d+1}) T(s_{d+1}|s_{d,i}, a) \, \mathrm{d}s_{d+1} \, \mathrm{d}o. \quad (24)$$

The analysis for the first two terms is already presented in [26]. We only rehearsal their arguments here. Observing that $\mathbb{E}[V_{d+1}^*(\tau(b, a, o))|b_d] = \int_{\mathcal{S}} \mathbf{V}_{d+1}^*(s_d, b_d, a) b_d \cdot \mathrm{d}s_d$, and $\mathbf{V}_{d+1}^*$ is bounded by $R_{\max}/(1 - \gamma)$, it follows that the first term is bounded by $\lambda$ with a probability at least $1 - \eta$. The second term can be considered as an estimator of

$$\mathbb{E}\left[ \mathbf{V}_{d+1}^*(s_d, b_d, a) - V_{d+1}^*(\tau(b_d, a, o)) \right] =$$

$$\int (\mathbf{V}_{d+1}^*(s_d, b_d, a) - V_{d+1}^*(\tau(b_d, a, o))) Z(o|a, s_{d+1}) T(s_{d+1}|s_d, a) b_d(s) \, \mathrm{d}s_{d:d+1} \, \mathrm{d}o = 0. \quad (25)$$

Since $\mathbf{V}_{d+1}^*(s_d, b_d, a) - V_{d+1}^*(\tau(b_d, a, o))$ is bounded by $2R_{\max}/(1 - \gamma)$, we can bound the second term by $\lambda/2$ with a probability of at least $1 - \eta$.

For the third term, it should be noted that only part of observation branches are expanded in AdaOPS. The value of a belief $\bar{b}$ that are not expanded is set to the value of the nearest belief $\bar{b}'$ in the belief packing. Therefore, for a belief $\tau(\bar{b}_d, a, o_i)$ in the packing, we have

$$\left| V_{d+1}^*(\tau(b_d, a, o_i)) - \tilde{V}_{d+1}^*(\bar{b}_d, a, o_i) \right| = \left| V_{d+1}^*(\tau(b_d, a, o_i)) - \hat{V}_{d+1}^*(\tau(\bar{b}_d, a, o_i)) \right| \quad (26)$$

$$\leq \epsilon_{d+1}.$$

For a belief $\tau(\bar{b}_d, a, o_i)$ not in the packing, there must exist an observation $o_j$ such that $\|\tau(\bar{b}_d, a, o_i) - \tau(\bar{b}_d, a, o_j)\|_1 \leq \delta$, and $\tau(\bar{b}_d, a, o_j)$ is in the packing. Then, we have

$$
\begin{aligned}
&\left| V_{d+1}^*(\tau(b_d, a, o_i)) - \tilde{V}_{d+1}^*(\bar{b}_d, a, o_i) \right| \\
&= \left| V_{d+1}^*(\tau(b_d, a, o_i)) - \hat{V}_{d+1}^*(\tau(\bar{b}_d, a, o_j)) \right| \\
&\leq \left| V_{d+1}^*(\tau(b_d, a, o_i)) - V_{d+1}^*(\tau(b_d, a, o_j)) \right| + \left| V_{d+1}^*(\tau(b_d, a, o_j)) - \hat{V}_{d+1}^*(\tau(\bar{b}_d, a, o_j)) \right| \\
&\leq \left| V_{d+1}^*(\tau(b_d, a, o_i)) - V_{d+1}^*(\tau(b_d, a, o_j)) \right| + \epsilon_{d+1}.
\end{aligned}
\tag{27}
$$

For any function $\alpha$ bounded by $\frac{R_{\max}}{1-\gamma}$, $|\int \alpha(s)b(s)\,\mathrm{d}s - \sum_{i=1}^N w_{d+1,i}\alpha(s_{d+1,i})|$ is bounded by $\lambda$ with a probability of at least $1 - \eta$. Therefore, we have the assumptions in Lemma 1 holding with a probability of at least $1 - 2\eta$. Consequently, for a belief $\tau(\bar{b}_d, a, o_i)$ not in the packing, the error $\left| V_{d+1}^*(\tau(b_d, a, o_i)) - \tilde{V}_{d+1}^*(\bar{b}_d, a, o_i) \right|$ is bounded by $2\lambda + \frac{R_{\max}}{1-\gamma}\delta + \epsilon_{d+1}$ with a probability of at least $1 - 2\eta$. Hence, we can bound the error of all child beliefs by the same quantity as well with probability $1 - 2N\eta$.

With the above argument, we have bounded (B) by $7\lambda/2 + \frac{R_{\max}}{1-\gamma}\delta + \epsilon_{d+1}$. When combined with the part (A), we get

$$
\begin{aligned}
|Q_d^*(b_d, a) - \hat{Q}_d^*(\bar{b}_d, a)| &\leq (1-\gamma)\lambda + \gamma\left(\frac{7}{2}\lambda + \frac{R_{\max}}{1-\gamma}\delta + \epsilon_{d+1}\right) \\
&= \lambda + \gamma\left(\frac{5}{2}\lambda + \frac{R_{\max}}{1-\gamma}\delta + \epsilon_{d+1}\right),
\end{aligned}
\tag{28}
$$

with a probability $1 - (3 + 2N)\eta$.

Notice that the action value estimation error is related the value estimation error in the following way:

$$
\begin{aligned}
|V_d^*(b_d) - \hat{V}_d^*(\bar{b}_d)| &= |\max_{a \in \mathcal{A}} Q_d^*(b_d, a) - \max_{a \in \mathcal{A}} \hat{Q}_d^*(\bar{b}_d, a)| \\
&\leq \max\left( |Q_d^*(b_d, a^*) - \hat{Q}_d^*(\bar{b}_d, a^*)|, |Q_d^*(b_d, \hat{a}^*) - \hat{Q}_d^*(\bar{b}_d, \hat{a}^*)| \right) \\
&\leq \lambda + \gamma\left(\frac{5}{2}\lambda + \frac{R_{\max}}{1-\gamma}\delta + \epsilon_{d+1}\right) = \epsilon_d,
\end{aligned}
\tag{29}
$$

where $a^* = \arg\max_{a \in \mathcal{A}} Q_d^*(b_d, a)$, and $\hat{a}^* = \arg\max_{a \in \mathcal{A}} \hat{Q}_d^*(\bar{b}_d, a)$. Expanding the recurrence, we find the error at the depth 0 is given by

$$
\begin{aligned}
|V_0^*(b_0) - \hat{V}_0^*(\bar{b}_0)| \leq \epsilon_0 &= \lambda + \gamma\left(\frac{5}{2}\lambda + \frac{R_{\max}}{1-\gamma}\delta + \epsilon_1\right) \\
&\leq \frac{\lambda}{1-\gamma} + \left(\frac{5}{2}\lambda + \frac{R_{\max}}{1-\gamma}\delta\right)\frac{\gamma}{1-\gamma} + \gamma^D\frac{2R_{\max}}{1-\gamma} \\
&= \frac{1 + \frac{5}{2}\gamma}{1-\gamma}\lambda + \frac{\gamma R_{\max}}{(1-\gamma)^2}\delta + \frac{2\gamma^D R_{\max}}{1-\gamma}.
\end{aligned}
$$

By setting $\delta = \frac{(1-\gamma)^2 \epsilon}{3\gamma R_{\max}}$, $\lambda = \frac{(1-\gamma)\epsilon}{3(1+5\gamma/2)}$, and $D = \log_\gamma \frac{(1-\gamma)\epsilon}{6R_{\max}}$, we can guarantee $|V_0^*(b_0) - \hat{V}_0^*(\bar{b}_0)| \leq \epsilon$ with a probability of at least $1 - (3 + 2N)|\mathcal{T}|\eta$. Here, $|\mathcal{T}|$ denotes the number of action nodes on the tree. We have $|\mathcal{T}| \leq |\mathcal{A}| \frac{(|\mathcal{A}| \min(N, P_{\max}^\delta))^D - 1}{|\mathcal{A}| \min(N, P_{\max}^\delta) - 1} \leq |\mathcal{A}|(|\mathcal{A}| \min(N, P_{\max}^\delta))^D$. Thus, the total failure probability is at most $5|\mathcal{A}|N(|\mathcal{A}| \min(N, P_{\max}^\delta))^D \eta$. $\qquad\square$

### C.3 Running Time Analysis

AdaOPS builds a tree of size at most $O((|\mathcal{A}| \min(P_{\max}^\delta, N_{\max}))^D)$, and it takes most of its time to expand new nodes. Expanding a leaf node with $m$ particles takes $O(|\mathcal{A}|m)$ time for propagating particles, $O(|\mathcal{A}|P_{\max}^\delta m)$ time for weighting new beliefs, $O(|\mathcal{A}|P_{\max}^\delta m^2)$ time for generating packings, and $O(|\mathcal{A}|P_{\max}^\delta m)$ time for initializing bounds. Although KLD-Sampling seems to be

resource-intensive due to the existence of a grid, it can be made efficient by not storing the entire grid but only the coordinate of samples. This way, the computing and storage overhead of KLD-Sampling is at an order of $O(mn_\mathcal{S} \log d_\mathcal{S})$, where $n_\mathcal{S}$ is the number of state dimensions, and $d_\mathcal{S}$ is the number of grids in each dimension.

It depends on the specific context to determine which of these is more costly. In domains with high dimensional observation space, weighting new beliefs dominates since the observation function $Z$ may take more time. In other cases, particle propagation and bounds initialization often achieve dominance.

# D  Experiments

The solvers used in our experiments are implemented by Zachary Sunberg and licensed under the MIT "Expat" License. The Roomba domain is attributed to [28], and we have obtained consent to cite it.

## D.1  Domains

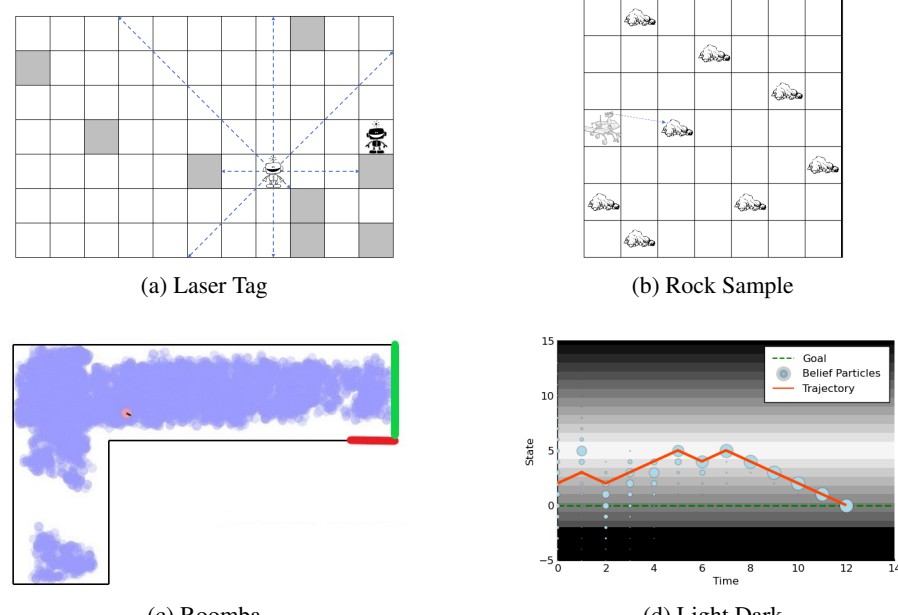

| (a) Laser Tag | (b) Rock Sample |

| (c) Roomba | (d) Light Dark |

Figure 4: Four domains: (a) Laser Tag. A robot (white) equipped with eight laser rangefinders tries to locate itself and tags an escaping target (black). (b) Rock Sample. A Mars rover estimates the quality of rocks by its noisy sensor, samples good rocks as many as possible, and then reaches the exit area (east boundary). (c) Roomba. A robotic vacuum cleaner (red circle) attempts to locate itself in a familiar room and reach the target region (green) while avoiding falling down the stair (red). Blue points visualize the belief at the current step. (d) Light Dark. To reach the origin, the agent moving on a real line needs to locate itself by going to the bright region. The red line shows the trajectory of the agent using AdaOPS, and the white circles represent the belief.

### D.1.1  Laser Tag

Laser Tag is an extended version of Tag, compared to which Laser Tag has a significantly larger observation space. At each timestep, the robot will get noisy discrete measurements from eight laser rangefinders, with which it will try to locate itself and the target. There are some obstacles on the map that lasers and the robot cannot pass through. The robot can move to the four adjacent positions, paying $-1$. If the escaping target is correctly tagged, it is rewarded $+10$. A penalty of $-10$ will be given if it tags wrongly.

### D.1.2 Rock Sample (RS)

Rock Sample is a common benchmark in the POMDP field. In a rock sample problem RS(n,m), a rover moves on an $n \times n$ grid world with $m$ rocks. Each rock could be either good or bad. The goal of the rover is to sample all the good rocks and exit by the east boundary. Each step, the rover can move to an adjacent grid, sense a rock, or sample a rock. Moving and sampling do not produce any observation. When executing a sensing action, the robot observes the rock's status with a noise that increases exponentially with the distance to the rock. Sampling will reward $+10$ for good rock, $-10$ for a bad one, and $0$ for nothing. Finally, the rover leaves the map from the east boundary and gets a reward of $+10$.

### D.1.3 Roomba

Roomba is a localization POMDP problem. A robotic vacuum cleaner, also known as Roomba, finds itself in a familiar room but does not know its exact position. It then tries to locate itself and enters the adjoining room (target area). When equipped with a Lidar sensor, it receives a noisy continuous measurement. With a Bumper sensor, it can only sense its collision, causing the localization extremely hard. For Lidar Roomba, a natural lower bound for the optimal value of belief $b$ is what we call Delayed MDP (DMDP), $\alpha^{\text{DMDP}}(s) = \sum_{i=0}^{n-1} \gamma^i r_{\text{time}} + \gamma^n \alpha^{\text{MDP}}(s)$, where $r_{\text{time}}$ is the time penalty. The term, Delayed MDP, means that the robot keeps rotating in situ for $n$ steps until its position is known with certainty and then operates in line with the optimal MDP policy. At each timestep, it receives a time penalty of $-0.1$. If the robot hits a wall, it gets a penalty of $-1$. It is rewarded $+10$ when reaching the target (green region) and is penalized $-10$ when falling down the stairs (red region).

### D.1.4 Light Dark

The Light Dark domain has a 1d continuous state that stands for the agent's position on a real line. The agent can move deterministically with action $+1$ or $-1$. The goal is to reach the origin. It can also take action $0$, which will reward $+10$ if the agent is close to the origin (within an error of 1) and $-10$ if not. Each movement has a cost of $-1$. To make certain its position, the agent needs to move towards the light region to obtain better vision. The agent is initially situated according to a normal distribution of $\mathcal{N}(2, 3)$, and the light region is located around 5. The observation is continuous and distributed according to $\mathcal{N}\left(\text{x}, \frac{|x-5|}{\sqrt{2}} + 0.01\right)$, where $x$ is the position of the agent.

## D.2 Hyperparameters and Heuristics

The ranges of grid search are shown in Table 2.

Table 2: Ranges for Hyperparameter Selection

| | |
|---|---|
| AdaOPS | $m_{\min} \in \{10, 30, 100\}$ $\delta \in \{0.1, 0.3, 1.0\}$ |
| DESPOT | $K \in \{30, 100, 300\}$ $\lambda \in \{0.0, 0.001, 0.01, 0.1\}$ |
| POMCPOW | $c \in \{1, 10, 100, 1000\}$ $\alpha_O \in \{0.01, 0.03, 0.1, 0.3, 1.0\}$ $k_O \in \{1.0, 2.0, 4.0, 8.0\}$ |

The hyperparameters selected for each algorithm are shown in Table 3. For AdaOPS without belief packing, $\delta$ is not presented since it is set to 0.

AdaOPS requires a state grid partitioning the state space into multidimensional bins. The size of the grid is determined such that it is an appropriate discretization of the state space that can faithfully reflect the dispersion of beliefs. Since a fine discretization results in enormous particle numbers according to Equation (2), in practice, we discretize the state space into bins of roughly a hundred. Laser Tag has a discrete state space of size $7 \times 11$ and needs no more additional discretization. Roomba has a continuous state space of size $40 \times 25$. We further discretize it into $16 \times 10$, as shown in Figure 5. Other reasonable discretization will yield similar results but having different optimal hyperparameters. For Light Dark, a natural choice is to discretize the state space (a real line) with

Table 3: Hyperparameters Selected

|  |  | Laser Tag | RS(15,15) | Bumper Roomba | Lidar Roomba | Light Dark |
|---|---|---|---|---|---|---|
| AdaOPS | $m_{\min}$ | 10 | 100 | 10 | 30 | 10 |
|  | $\delta$ | 0.1 | 0.1 | 0.0 | 0.3 | 1.0 |
| AdaOPS(AR) | $m_{\min}$ | 30 | 100 | 10 | 30 | 30 |
|  | $\delta$ | 0.3 | 0.1 | 0.0 | 0.3 | 1.0 |
| AdaOPS(KLD-S) | $m_{\min}$ | 100 | \ | 10 | 30 | 100 |
|  | $\delta$ | 0.1 |  | 0.0 | 0.3 | 1.0 |
| AdaOPS(BP) | $m_{\min}$ | 30 | 100 | \ | 10 | 10 |
| DESPOT | $K$ | 300 | 100 | 30 | 30 | 30 |
|  | $\lambda$ | 0.01 | 0.0 | 0.1 | 0.01 | 0.1 |
| POMCPOW | $c$ | 10 | 10 | 100 | 1000 | 10 |
|  | $\alpha_O$ | 0.03 | 1.0 | 1.0 | 0.03 | 0.03 |
|  | $k_O$ | 4.0 | 1.0 | 1.0 | 2.0 | 4.0 |

the step size 1. Rock Sample needs no state grid, as the dispersion of belief does not change. The heuristics adopted in different domains are demonstrated in Table 4. The fixed action policy for Rock Sample is moving east since the rover receives $+10$ whenever it arrives at the exit (east boundary).

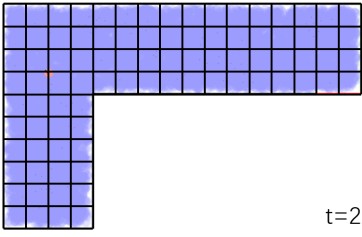 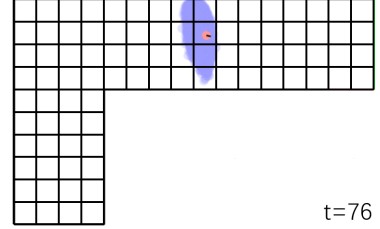

Figure 5: Roomba Grid

The heuristics in different domains are shown in Table 4. It should be noted that although the QMDP upper bound can achieve better performance on the Rock Sample domain, AdaOPS and DESPOT consume more memory. Therefore, we use MDP upper bound instead of QMDP upper bound.

Table 4: Heuristics in Different Domains

|  | Lower Bound | Upper Bound | Value Estimator |
|---|---|---|---|
| Laser Tag | Blind Policy | QMDP | MDP |
| Rock Sample | Fixed Action Policy | MDP | MDP |
| Lidar Roomba | Delayed MDP | MDP | MDP |
| Bumper Roomba | Blind Policy | QMDP | MDP |
| Light Dark | Random Policy | MDP | MDP |

### D.3 Time Sensitivity Analysis

This section provides a time sensitivity analysis in Light Dark, in which we adjust the planning time to $t \in \{0.125s, 0.25s, 0.5s, 1s, 2s, 4s, 8s\}$ and test their performance for $1,000$ episodes. As shown in Figure 6, even when the planning time is short, AdaOPS still outperforms POMCPOW and DESPOT. Besides, the performance of AdaOPS is more stable across all settings (with smaller SEM) compared to the others.

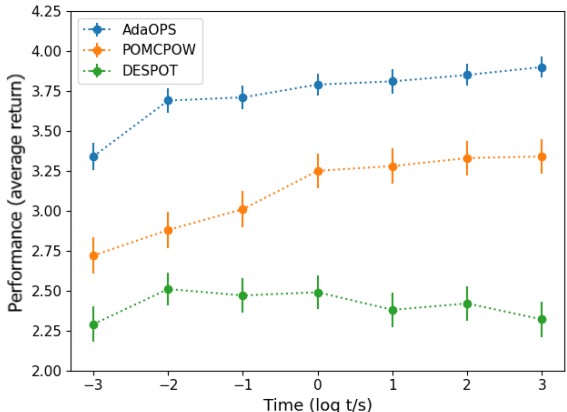

Figure 6: Time sensitivity analysis in Light Dark. Each point represents the average return for 1000 trials of the solver given a specific planning time. The vertical line on the dot represents the standard error mean (SEM).

## D.4  Hyperparameter Sensitivity Analysis

This section provides a hyperparameter sensitivity analysis in Light Dark, in which we change the hyperparameters $\delta$ and $m_{\min}$ and test their performance for $1,000$ episodes. As illustrated in Figure 7, AdaOPS is relatively stable to hyperparameter changing, and half tests achieve an average discounted return greater than 3.

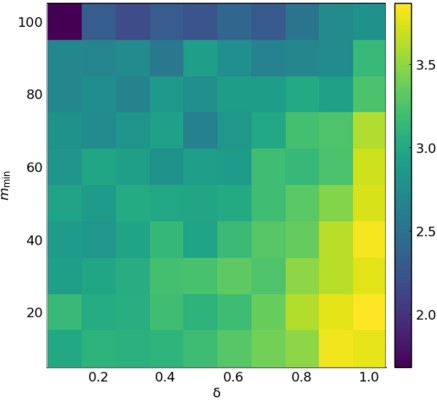

Figure 7: Hyperparameter sensitivity analysis in Light Dark. The color denotes the average discounted return of $1,000$ episodes, which varies slowly with respect to the changing of hyperparameters $\delta$ and $m_{\min}$.

## D.5  Tree Building Analysis

Figure 8 illustrates the exploration depths, particle numbers, and observation numbers throughout an episode in various domains. In most domains, the state uncertainty diminishes with more information gathered, and the belief gradually concentrates on a small region with the particle number, given by KLD-Sampling, declining. Since approximating a belief requires less particles, the algorithm can expand more nodes and make a deeper tree search. Besides, a low state uncertainty also induces a smaller belief packing, which is for two reasons. On one hand, the support of the observation distribution is affected by the support of the belief $b$, i.e., $\sup(\Pr(o \mid b, a)) = \bigcup_{s \in \sup(b)} \sup(Z(o \mid s, a))$ for action $a$. On the other hand, it is often hard to distinguish between the remaining possible states. If otherwise, these states may have already been made impossible by previous observations. Notice

that the depth of exploration decreases at the end of each episode. It is because AdaOPS already finds the optimal policy, and there is no need to search deeper.

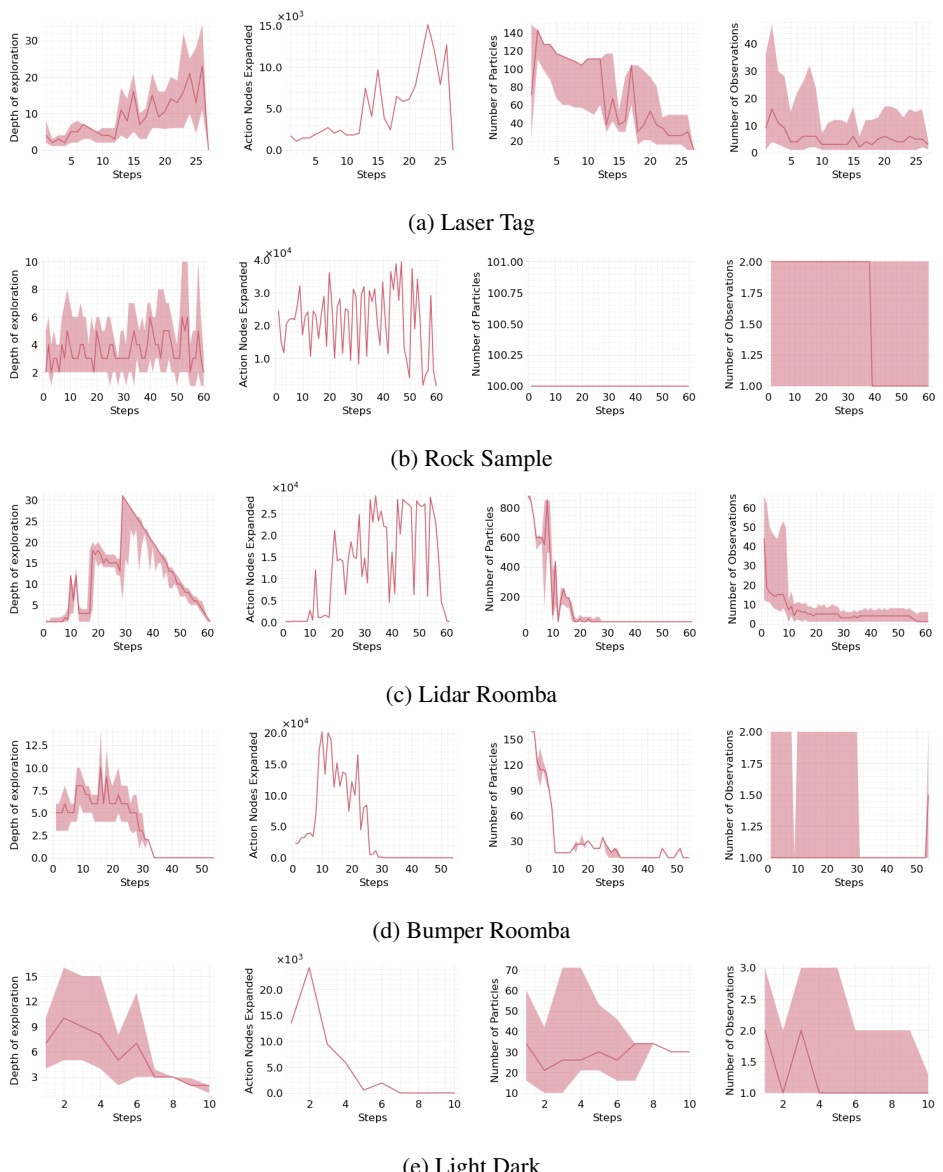

(a) Laser Tag

(b) Rock Sample

(c) Lidar Roomba

(d) Bumper Roomba

(e) Light Dark

Figure 8: Tree building analysis for AdaOPS in different domains. The solid red line denotes the median, and the shadow area denotes the 90% confidence interval.