# OpenReview forum: "Adaptive Online Packing-guided Search for POMDPs"
_NeurIPS.cc/2021/Conference — NeurIPS 2021 Poster_

### Official Review · Reviewer_rLs9 · 2021-06-29

**Rating:** 6
**Confidence:** 4

**Summary:**

This paper proposes an online tree-search algorithm for partially observable Markov decision processes (POMDPs). Taking sampling-based tree search style (e.g., POMCP) approach as a basis, the paper proposes to employ an adaptive particle filter that tunes the number of samples in each branch of the tree using the KLD-sampling method (Fox, 2001). To prune the search tree, the paper proposes a new method of observation aggregation called belief packing: here, observation branches resulting in similar beliefs (and hence similar values) are aggregated together. The benefits of these ideas applied to online tree search planning are demonstrated empirically.

**Limitations And Societal Impact:**

Yes, this is adequately addressed.

**Main Review:**

Strengths:

- The idea of belief packing is an interesting and reasonable alternative to progressive widening and other strategies used to deal with very large observation spaces. In the context of online tree search for POMDPs, this seems to be a novel contribution.
- The empirical results nicely show cases where it is helpful and where it does not bring additional benefits. The results overall show improvements over DESPOT and POMCPOW, indicating the usefulness of the proposed approach where the requirements of the proposed method can be fulfilled.

Limitations:

1. The main idea is reasonable and seems useful, but is still a modification of the well-known online tree search approach to solving POMDPs. The contribution might be relevant for POMDP practicioners, but may not receive much interest elsewhere.

2. One algorithmic limitation is that to perform resampling, the state space has to be partitioned into a grid. This is unlikely to scale to high dimensional state spaces. Another limitation is the requirement of access to a fully specified observation model for particle reweighting. This makes the setting where the proposed algorithm can be applied more limited than DESPOT and POMCP.

3. The belief packing idea is nice, but it does not seem to be completely developed in the paper. For instance, I was expecting the delta-packing number to make an appearance in the main theorems of the paper. I was not sure if the delta there is to be interpreted as the packing number, or if the two theorems are generic results for particle-based online tree search.

4. I do not appreciate the choice to defer discussion of related work to an appendix. I think this is very useful information to understand the relation of this work to others, and should be included in the main paper.

5. Some parts of the paper are unclear, and the two theorems rely heavily on [17], also for the underlying asumptions, which makes the paper not that self-contained.

I would appreciate comments from the authors especially on points 2 and 3.

Detailed comments:

- The writing in the paper could be improved to make it more accessible:
* The introduction could motivate the need for the algorithm generally at least a little, instead of the very focused technical POMDP angle taken.
*Especially Sect. 3 needs improvement, it introduces many concepts and includes many forward references that make following the flow of ideas arduous. The basic algorithm seems very similar to POMCP, although with several modifications, so it might even make sense to explicitly note the differences to POMCP for improved clarity.
* Also in Sect 3., it would be worthwhile to focus on the main contributions of this paper, and give less emphasis on already known algorithmic ideas.

- Is the Light Dark domain the same as in Table 1 of Sunberg & Kochenderfer (ICAPS 2018)? Or has the reward function been scaled somehow? I am wondering since the reported average performance is quite different.

** The following comments are for the information of the authors only, and do not need to be answered.

- The belief packing idea is a type of observation aggregation into meta observations considered in earlier work, e.g., [2, 3]. Expanding on this connection might give a nice alternative view of belief packing.

Minor:
Line 22-23: Exact belief updating is possible with parametric distributions in certain continuous-state cases, see, e.g., Kalman filters in linear-Gaussian POMDPs.
Line 25-26: Papadimitriou & Tsitsiklis only consider the finite horizon undiscounted expected cost case for POMDP. For the infinite horizon case, see [1].
Line 65: "average" --> "expected sum of"?
Line 69: Bayesian filtering is not necessarily approximate, it can also be exact.
Eq. (2): Compared to Fox (2001), this equation is missing the power of three for the term in the parentheses. Same mistake later on Line 208.

References:
[1] Madani et al., "On the undecidability of probabilistic planning and related stochastic optimization problems", Artificial Intelligence, Volume 147, Issues 1–2, July 2003, Pages 5-34
[2] Hoey & Poupart, "Solving POMDPs with Continuous or Large Discrete Observation Spaces", IJCAI 2005
[3] Porta et al., "Point-Based Value Iteration for Continuous POMDPs", JMLR, Vol. 7, 2329-2367

Comments added after author response:
Thanks for answering my comments. While I remain by my original rating, I hope to see a future version of the paper incorporating useful suggestions from all the reviews. Besides an updated main theorem, an empirical demonstration of how to deal with scalability concerns would be very helpful.

**Time Spent Reviewing:**

3

---

> ### Author Response · Authors · 2021-08-09
> **Response**
>
> Thank you for the invaluable comments. We will answer your concerns piece by piece.
>
> **On point 1**, the belief approximation and the treatment of large observation space in POMDP online planning are two important topics and hinder POMDPs from application in real-world problems. We believe this will receive growing attention in the future.
>
> **On point 2**:
> - On the scalability, there are several ways for scaling up to high-dimensional problems. First, as mentioned in [Fox, 2001], the bins can be efficiently implemented as tree structures, where each internal node on the tree representing a separation plane. Second, instead of partitioning the entire state space into a grid, we could pick a few dimensions which are enough for representing the state uncertainty. For example, in the Roomba domain, the state has three dimensions, x coordinates, y coordinates, and orientations. Nevertheless, only coordinates are discretized for measuring the belief dispersion (see Appendix F.2 for details). Third, we can use random projection techniques for dimension reduction, such that distances are approximately preserved. With these methods, we can scale the KLD-Sampling method to handle most existing POMDP problems.
> - For high-dimensional continuous observation problems, we believe an observation model is necessary. Otherwise, it will not be possible to perform, even approximately, a belief update since discretization will not work in high-dimensional problems. Besides, the demand for an observation model is actually not a particularly strong requirement. Since the generative model implicitly specifies the observation distribution, we can use machine learning techniques to retrieve an approximate observation model from it.
>
> **On point 3**, the first theorem is quite a generic one and can be easily adapted to special cases. Actually, when specialized to AdaOPS, it is possible to include an additional dependency on the delta-packing number. This will result in an improved bound which relaxes the requirement $3|\mathcal{A}| (3|\mathcal{A}|N)^D\exp(-N⋅t_{\max})≤\eta$ in Theorem 1 to be $3|\mathcal{A}| (3|\mathcal{A}|\min(P_{\max}^\delta,N))^D\exp(-N⋅t_{\max})≤\eta$. This improvement follows from that the maximum packing number restricts maximum tree size, which is used for bounding the total failure rate with union bound. Since there is another $N$ in the exponent, this improvement is dominated and will not affect the order of $N$. Thus, we dropped this dependency in exchange for simplicity. However, we agree on the significance of this dependency in showing how the packing number affects planning and will add it back in the final version.
>
> **On point 4**, we put the related work in the appendix due to the space constraint. We will add this section to the main text in the final version.
>
> **On point 5**, we cannot shake off the dependency on [17] for now because an alternative proof factoring in resampling would require an exponential concentration inequality of particle filters for all action and observation sequences, which, to the best of our knowledge, does not exist now. We are still working actively on this problem.
>
> We are grateful for the detailed comments and will refine our work accordingly to improve clarity. The Light Dark domain we used is different from that in [Sunberg et al., 2018]. This one we used has a continuous state, whereas that in [Sunberg et al., 2018] is discrete. Moreover, their domain has additional actions of large moving strides, $+10$ and $-10$, and broader initial distribution. Since it is not evident which is more difficult, we tested AdaOPS on their domain. It turns out AdaOPS achieves the best performance, $62.77\pm0.37$, with $m_{\min}=30$ and $\delta=0.3$.
>
> Thanks again for your careful comments.
>
> *Fox, Dieter. "KLD-Sampling: Adaptive Particle Filters." In Advances in Neural Information Processing Systems, Vol. 14, 2001.*

---

### Official Review · Reviewer_g6M6 · 2021-07-16

**Rating:** 7
**Confidence:** 4

**Summary:**

This paper presents an online planning algorithm, called Adaptive Online Packing-guided Search (AdaOPS), for Partially Observable Markov Decision Processes.
The article addresses the problem of improving the approximations of a belief using a particle filter. This improvement comes from two elements: adaptive particle filtering and belief packing.

Specifically, During the planning step, a particle resampling procedure is performed adaptively on new leaf nodes.
This happens when the number of particles in the belief of the node is greater than twice the Effective Sample Size (ESS), as explained in (Kish [11], Liu [12]).
AdaOPS use a KLD-sampling for this adaptive resampling (Fox [13], Li, Sun, Satter [15]), this guarantees a good approximation using a limited number of particles.

The article also presents a technique called Belief Packing, inspired by (Zhang, Hsu, Lee [17]).
It is used to fuse similar beliefs, thus the planning procedure explores only beliefs that are significantly different.
This improves the scalability of the online algorithm.

The paper shows that with enough planning time, this packing produces results that are $\epsilon$-optimal (i.e., the difference between the original belief and the belief that employs the packing is lower than a constant $\epsilon$ with a probability of at least $1 - \eta$).
The article also presents an empirical evaluation of AdaOPS in four different domains (Laser Tag, Rocksampe, Roomba, and Light-Dark). The results show that AdaOPS can outperform state of the art planning algorithms (POMCPOW, ARDESPOT).

**Limitations And Societal Impact:**

Authors indicate that they do not see potential negative societal impacts with their work and I agree.

**Main Review:**

The paper addresses an important problem, namely, improving the approximation of a correct belief using particle filters and this is achieved by combining different techniques.
Each technique, by itself, does not provide an original contribution, but the non-trivial combination of such techniques is an interesting and valuable contribution.
In this regard, I believe that the paper should discuss in more detail why combining adaptive particle filters and belief packing is beneficial with respect to considering the two techniques independently.

The main theoretical proof presented in the article concerns the convergence to an optimal policy given enough planning time.
The proof is, as far as I understand, correct, but it is not clear to me how fast this convergence is (i.e., is this result useful in practical applications ?).

The empirical evaluation section considers four domains and presents a comparison of the performance achieved by AdaOPS with two state-of-the-art planning algorithms. In each instance, solvers have 1 second for planning a step.
I find this experimental setup incomplete.
The main theoretical proof presented in the article concerns the convergence to an optimal solution given enough time, but the impact of this result is not explored empirically.
In particular, it is not clear how fast and reliable this convergence is.
Using 1 second for step is a reasonable choice, but it should not be the only one considered.
It could be that AdaOPS performs very well with constrained resources but it is outperformed by other algorithms when the planning time is shorter or longer. This may still be a valuable result for AdaOPS but should be clearly discussed to better define in which situation it is beneficial to use AdaOPS.
I point this out because the performance achieved by POMCP in Rocksample in this scenario seems low compared to my expectation. I believe that with more resources POMCP can achieve a greater reward, is AdaOPS still significantly better in other cases?

The paper is well written, but I find the usage of the appendix excessive. This makes the paper hard to follow. In particular, the main paper should present a brief description of the proofs for theorems 1 and 2. Similarly, the short discussion of the size of the generated tree should be moved from section 6 to section 4.


**Time Spent Reviewing:**

30

---

> ### Author Response · Authors · 2021-08-09
> **Response**
>
> Thanks for your careful reading and prized comments. Your concerns are explained point by point.
>
> **Q1**: Why is it beneficial to combine adaptive particle filters and belief packing with respect to considering the two techniques independently?
>
> **A1**: We agree that the benefits of combining two techniques should be made clear. Belief packing and adaptive particle filters solve orthogonal problems yet work reciprocally. Without an accurate belief approximation using the adaptive particle filter, the distance estimation between sibling beliefs will not be accurate. It is especially the case when we use SIS techniques for belief approximation, where most particles have negligible weights, and the estimated L1 distance will be inaccurate. Without belief packing, we will not be able to handle a large number of observations. It will be impractical to perform belief updating with particle filters for numerous observations. We will add more discussion about the benefits in the main text in the final version upon acceptance.
>
> **Q2**: How fast is the convergence? Is it useful in practical application?
>
> **A2**: AdaOPS has a convergence rate of $O\left(\frac{\ln N}{N}\right)$, derived by reformulating Theorems 1 and 2. However, this result is not helpful in real application because online planning, in the worst case, requires exponential time in searching depth $D$ (see Appendix E.3), which is the case even for the MDP counterpart [Kearns et al., 2002]. In most non-trivial problems, we cannot wait for the algorithm to terminate. Nonetheless, the convergence will still have a fundamental implication for the finite-time performance, respecting which AdaOPS is demonstrated to be superior.
>
> **Q3**: How does AdaOPS perform when the planning time is shorter or longer?
>
> **A3**: Since the required time for online planning increases exponentially with search depth, online planning algorithms are insensitive to the planning time. We conducted another set of experiments on RockSample(15,15) and LightDark, setting the planning time as 0.5, 2.0, and 4.0 seconds with all other hyperparameters unchanged. The performance scores (higher is better) are listed as follows.
>
> **RockSample(15,15)**
>
> | Time | AdaOPS | DESPOT | POMCPOW |
> | :--- | :---: | :---: | :---: |
> | 0.5 | **16.73±0.31** | 14.14±0.31 | 9.32±0.31 |
> | 2.0 | **16.92±0.32** | 15.44±0.32 | 9.50±0.30 |
> | 4.0 | **17.05±0.32** | 15.80±0.32 | 9.84±0.31 |
>
> **LightDark**
>
> | Time | AdaOPS | DESPOT | POMCPOW |
> | :--- | :---: | :---: | :---: |
> | 0.25 | **3.63±0.07** | 2.33±0.09 | 2.99±0.11 |
> | 0.5 | **3.72±0.07** | 2.57±0.09 | 3.38±0.10 |
> | 2.0 | **3.83±0.07** | 2.51±0.09 | 2.96±0.11 |
> | 4.0 | **3.85±0.07** | 2.37±0.09 | 3.16±0.11 |
>
> Noticeably, both POMCPOW and DESPOT have better performance given 0.5s than given 1.0s in LightDark. Hence, we added another experiment using 0.25s in Light Dark, and their performance drops. This result demonstrates that the performance of AdaOPS does not degrade too much with less planning time. Moreover, DESPOT and POMCPOW cannot match the performance of AdaOPS with eight times more computing resources.
>
> **Q4**: When is it beneficial to use AdaOPS?
>
> **A4**: AdaOPS improves over two fundamental integral approximations in Equation (4). Therefore, it would always be at least a reasonable choice to use AdaOPS. Especially in domains with high state uncertainty, AdaOPS should be the first choice for handling the uncertainty better.
>
> **Q5**: Why is the performance of POMCP below expectation?
>
> **A5**: The performance of POMCP in the Rock Sample is below expectation because Rock Sample is a domain in which the optimal value varies a lot with respect to different maps. There exist specific maps where POMCPOW could achieve an expected discounted return over 20. Our results are averaged over one hundred randomly generated maps, which causes the performance of POMCP to be lower than that in other work.
>
> Concerning the writing, we will make every effort to make this paper more accessible following reviewers' comments. Thanks again for your thoughtful advice.
>
> *Kearns, Michael, Yishay Mansour, and Andrew Y. Ng. "A Sparse Sampling Algorithm for Near-Optimal Planning in Large Markov Decision Processes." Machine Learning 49, no. 2 (November 1, 2002): 193–208.*

---

> > ### Comment · Reviewer_g6M6 · 2021-08-25
> > **Comments on your response and further clarification questions**
> >
> > I would like to thank the authors for addressing my questions.
> >
> > The comment on why it is important to combine the two aspects of AdaOPS, the convergence rate, and the other experiments on RockSample and LightDark are convincing. In particular, I find the experiments run using [0.25, 8] seconds per step particularly important.
> >
> > I run further experiments using the provided code and specifically AdaOPS and POMCP on RockSample to get a better understanding of the difference between the two algorithms. Based on these experiments I agree with the authors that the map configuration can greatly influence the expected return.
> >
> > Given the responses and these results, I am willing to increase my rating from 5 to 6.
> >
> > However I still have a couple of doubts that authors should address:
> >
> > -) One of the things that I noticed running these experiments is that AdaOPS uses a lot more memory than POMCP. In RockSample(15, 15), AdaOPS used more than 16 GB of RAM, vs. POMCP that used ~600 MB of RAM. Is this a limitation of the current implementation of AdaOPS or is this due to a higher memory requirement of the method ? I cannot pinpoint the reason by reading the article. I think this is an important point that should be somehow addressed by the authors because memory can quickly become a bottleneck when  scaling to larger problems.
> >
> > -) I would like further clarifications for Q1. I understand why it is beneficial to use an adaptive particle filter to improve the approximation of the belief and thus the belief packing. However, it is less clear to me why the belief packing should reciprocally improve the adaptive particle filter aspect.

---

> > > ### Author Response · Authors · 2021-08-28
> > > **Response**
> > >
> > > Thanks a lot for reevaluating our work.
> > >
> > > Concerning the memory issue, AdaOPS and DESPOT consume more memory because they initialize the upper bound using the QMDP solution. The QMDP solver currently requires a large amount of memory, partly due to an implementation issue. The code attached below fixes this making it three times more memory-efficient, but indeed QMDP may fail to scale to larger problems. Therefore, we conduct another set of experiments in $\textrm{RockSample}(15,15)$ initializing the upper bound with MDP solution and fixed value $V_{\max}$, respectively. Besides, we benchmarked the memory allocation per step. As illustrated below, AdaOPS still consistently outperforms other algorithms while occupying memories of the same magnitude.
> > >
> > > | | Average Return | Memory Allocation (MB) |
> > > | :--- | :---: | :---: |
> > > | AdaOPS (MDP) | 17.29±0.21 | 401 |
> > > | AdaOPS (Fixed) | 10.68±0.20 | 450 |
> > > | DESPOT (MDP) | 15.60±0.21 | 409 |
> > > | DESPOT (Fixed) | 10.03±0.18 | 566 |
> > > | POMCP (MDP) | 10.40±0.18 | 94 |
> > > | POMCP (FixedActionPolicy) | 10.04±0.19 | 88 |
> > >
> > > It is noticeable that POMCP using the MDP solution as heuristics actually performs slightly better than that using fixed action policy, the one we used in the paper, which is unexpected and is presumably because the latter performs better due to stochasticity during the hyper-parameter selection phase. We will correct this in the final version.
> > >
> > > Regarding Q1, it should be noted that the particle filter technique is pricey since it needs to maintain a collection of particles for all beliefs, impairing its scalability. In practice, POMCPOW and DESPOT typically search deeper than AdaOPS does, as most beliefs in these methods comprise merely a few particles. While the particle filter ensures belief approximation quality, shallow searches will also lead to inferior performance. The belief packing boosts its scalability by pruning similar observation branches enabling it to have a deeper search.
> > >
> > > The following function is reimplemented, making the QMDP solver more memory-efficient.
> > > ```julia
> > > function RockSample.state_from_index(pomdp::RockSamplePOMDP{K}, si::Int) where K
> > >     if si == length(pomdp)
> > >         return pomdp.terminal_state
> > >     end
> > >     s = Vector{Int}(undef, K+2)
> > >     si -= 1
> > >     si, r = divrem(si, pomdp.map_size[1])
> > >     s[1] = r + 1
> > >     si, r = divrem(si, pomdp.map_size[2])
> > >     s[2] = r + 1
> > >     for i in 3:K+1
> > >         si, r = divrem(si, 2)
> > >         s[i] = r
> > >     end
> > >     s[K+2] = si
> > >     return RSState{K}(s[1:2], s[3:K+2])
> > > end
> > > ```

---

> > > > ### Comment · Reviewer_g6M6 · 2021-09-01
> > > > **comments on response**
> > > >
> > > > Thank you very much for the response this is of great help to clarify the memory issue.
> > > > I would strongly suggest authors to include in a future version of the paper a discussion on memory limitations due to the use of QMDP for large problems and to include (maybe in the sup. mat.) these new results.

---

### Official Review · Reviewer_7fyM · 2021-07-18

**Rating:** 5
**Confidence:** 4

**Summary:**

This paper proposes an online POMDP algorithm that performs belief tree search to find an action for current belief at each step. Building on the HSVI algorithm and more recent works using Monte Carlo belief tree approximations, the paper leverages on two ideas to achieve better performance: KLD sampling for more accurate belief representation, and the idea of belief packing to combine observation branches. The algorithm seems to have strong performance on several benchmark problems. A theoretical analysis is given.

**Limitations And Societal Impact:**

The theoretical analysis seem to have a few limitations that are not discussed. See the main review.

**Main Review:**

Belief representation and control of observation branching factor are two important considerations in the design of belief-tree based POMDP solver's performance. This paper proposes leveraging on the ideas of KLD-sampling and belief packing, and demonstrates that these two ideas work well on several benchmark datasets.

The paper is generally well-written and easy to follow, but discussion on related works should be in the main text rather than in the supplementary materials. The algorithm is an interesting combination of several ideas, which seem to be sound and effective in practice.

A theoretical analysis is given for a reduced version of the algorithm (without resampling), but it seems that there are a few limitations that are not discussed.  Theorem 1 shows that if two estimated beliefs are similar, then the estimated value for one is a good approximation to the true value of the other, provided that each estimated belief is represented by sufficiently many particles.
- It'd be helpful to discuss the magnitude of the sampel size $N$. Currently $N$ is related to other parameters in a complex inequality, and it'd be helpful to show $N$ grows as other parameters grow.
- It will be helpful to discuss on both the definition and the magnitude of $d_{\infty}^{\max}$, which is defined as the "essential supremum of the Rényi divergences between the sampling and target distributions". The "sampling and target distributions" seem to refer to the estimated beliefs and true beliefs. If that's the case, the analysis seems to focus on the discrete case only otherwise the Rényi divergences will be undefined. In addition, it seems $d_{\infty}^{\max}$ can be bounded in terms of the transition model and the observation model, and some discussion on the magnitude of $d_{\infty}^{\max}$ will help understanding the required sample size for good approximation.
- It is also not clear whether the analysis takes the errors due to belief packing into account. The first term in the bound seems to take the belief packing error at the current belief into account, but errors arising due to belief packing for descendant belief nodes doesn't seem to captured anywhere.
- The notation $\hat{V}^{*}$ isn't defined anywhere, but it seems to be the same as $V^{*}_{\text{AdaOPS}}$ in Theorem 2, which denotes the estimated optimal value given by AdaOPS.
- Theorem 2 sets $N$ to $m_{\min}$, but $m_{\min}$ is a hyperparameter that doesn't depend on the required value function error and the confidence level. How is it possible to have such a universally good sample size?

Overall, the proposed algorithm has some interesting ideas and seem to have good empirical performance, but it seems that the theoretical analysis has a few limitations.

**Time Spent Reviewing:**

4

---

> ### Author Response · Authors · 2021-08-09
> **Response**
>
> Thank you for the valuable comments.
>
> The related work is positioned in supplementary materials due to the limited space. We will move the related works to the main text upon acceptance.
>
> In what follows, we will address concerns raised on the theoretical part.
>
> **Q1**: What is the magnitude of the sample size $N$?
>
> **A1**: From the inequality in Theorem 1, it follows that the sample size $N$ is of an order of $O\left(\frac{R_{\max}d_{\infty}^{\max}D}{\lambda(1-\gamma)}\ln\frac{|\mathcal{A}|}{\eta}\right)$. We will add this in the final version.
>
> **Q2**: What are the definition and the magnitude of $d^{\max}_{\infty}$?
>
> **A2**: Formally, the $d^{\max}_{\infty}$ is the minimum value ensuring
>
> $$d_{\infty}^{\max}\geq d_{\infty}(\mathcal{P}^d || \mathcal{Q}^d)=\operatorname{ess}\sup_{x\sim\mathcal{Q}^d}\frac{\mathcal{P}^d(x)}{\mathcal{Q}^d(x)}$$
>
> to hold for all action and observation sequences of length $0,1,\ldots,D-1$, where $\mathcal{P}^d(s_{0:d})=\mu b_0(s_0)\prod_{n=1}^{d}T(s_{n}|s_{n-1},a_{n-1})Z(o_n|a_{n-1},s_{n})$, $\mu$ is a normalizing constant, and $\mathcal{Q}^d(s_{0:d})=b_0(s_0)\prod_{n=1}^{d}T(s_{n}|s_{n-1},a_{n-1})$. This term increases exponentially with respect to the depth $D$, which accounts for the exponentially increasing variance of the sequential importance sampling, as also mentioned in line 35. In the optimal filtering problem, it is well-known that we can establish a time uniform convergence for SIR particle filters under the assumption of exponential forgetting of the initial condition [Crisan et al., 2002]. If the same result holds for planning, we could prove an exponential improvement over SIS methods, on which we are now actively working.
>
> **Q3**: Are the errors from descendant beliefs captured?
>
> **A3**: The errors from descendant belief nodes are captured in Theorem 2. Briefly speaking, we bound the error recursively from leaves to the root. At each level, the error breaks down into two parts, error from descendant beliefs and the belief packing error at the current belief. See Appendix E.2 for detailed proof.
>
> **Q4**: Is $\hat{V}^*_{\text{AdaOPS}}$ defined?
>
> **A4**: We appreciate that the reviewer points it out. It is currently defined in Appendix E.1, standing for the value given by AdaOPS when terminating with the upper bound at root equal to its lower bound. Algorithm 2 also leaves out a terminating condition, which is to terminate when EU(bel) equals zero. We will correct these in the final version.
>
> **Q5**: How is it possible for $m_{\min}$ not to depend on the required error and confidence level?
>
> **A5**: When we set $N$ to $m_{\min}$, $m_{\min}$ should satisfy the requirements on $N$ in Theorem 1. This way, $m_{\min}$ depends on the required error and confidence level, i.e., $3|\mathcal{A}|(3|\mathcal{A}| m_{\min})^{D} \exp \left(-m_{\min} \cdot t_{\max }\right) \leq \eta$, where
> $$t_{\max }=\frac{\lambda(1-\gamma)}{3 R_{\max } d_{\infty}^{\max }}-\frac{1}{\sqrt{m_{\min}}}>0.$$
>
> *Crisan, D., and A. Doucet. "A Survey of Convergence Results on Particle Filtering Methods for Practitioners." IEEE Transactions on Signal Processing 50, no. 3 (March 2002): 736–46.*

---

> > ### Comment · Reviewer_7fyM · 2021-09-01
> > **theoretical analysis**
> >
> > Thanks for your anwers. I've taken a closer look at the theoretical analysis and have a few follow-up questions.
> >
> > Does the analysis only consider discrete states and observations? If the states and observations can be continuous, then it seems $d_{\infty}^{\max}$ can be infinite, and the theorems will not hold any more.
> >
> > The proof of Theorem 1 uses Appendix Theorem 1, but it seems Appendix Theorem 1 is from a paper which does not use belief packing to construct an approximate belief tree, while Theorem 1 analyzes an algorithm that uses belief packing, so it seems Appendix Theorem 1 is not applicable here. How is $\hat{V}^{\star}_{d}$ related to $\hat{V}^{\star}_{\text{AdaOPS}}$?
> >
> > In the proof of Theorem 2, $\epsilon_{d}$ is not defined. Does the "estimation error" of a belief at line 108 refer to $|V^{\star}(b) - \hat{V}^{\star}_{\text{AdaOPS}}(\hat{b})|$? Does each inequality below line 110 hold with probability $1 - \eta$ or all such inequalities hold simultaneously with probablity $1-\eta$?
> >
> > It'd be helpful to clearly define all the notations, and explicitly state all the assumptions, either in the paper/appendix.

---

> > > ### Author Response · Authors · 2021-09-02
> > > **Reply**
> > >
> > >
> > > Thanks for your comments. We answer your questions one by one as follows.
> > >
> > > **Q1**: Does the analysis only consider discrete states and observations? Could $d_{\infty}^{\max}$ be infinite?
> > >
> > > **A1**: The analysis does consider continuous states and observations. In fact, it is built assuming continuous state and observation spaces, though we can extend it to discrete spaces easily. Besides, $d_{\infty}^{\max}$ is assumed to be finite, which is reasonable because the importance weight can be unbounded if otherwise. These assumptions are introduced by [Lim et al., 2020] and are mentioned but not explicitly listed in Theorem 1 in our paper. Other assumptions in [Lim et al., 2020] are mostly regular, including that the action space is a finite set, the reward function is bounded, the horizon is finite, and we have access to a generative model $G$ and an observation function $Z$.
> > >
> > > **Q2**: Is Appendix Theorem 1 applicable here? How is $\hat V_d^*$ related to $\hat{V}^*_{\text{AdaOPS}}$?
> > >
> > > **A2**: Yes. We can couple the tree constructing process of two algorithms by first constructing a tree without belief packing as in [Lim et al., 2020] and then pruning similar branches with belief packing. Without belief packing, Appendix Theorem 1 is directly applicable ensuring an accurate $\hat V_d^*$ concentrated around the true value with high probability. Then, we factor in the extra errors introduced by belief packing and get $\hat{V}^*_{\text{AdaOPS}}$. This analysis is coarse since it does not consider the potential gain of belief packing in reducing the number of particles needed. However, a finer analysis will only tighten the bound by a logarithmic term as the $N$ in the exponent dominates. This gives rise to another question if the belief packing is beneficial. The main benefit of belief packing is that it reduces the tree size from $O(|\mathcal{A}|N)^D$ to $O((|\mathcal{A}|P_{\max}^\delta)^D)$. When $N$ is exponential with respect to $D$, the former will be of EXPSPACE complexity, as opposed to the PSPACE complexity suggested by the latter.
> > >
> > > **Q3**: What do the $\epsilon_d$ and "estimation error" refer to?
> > >
> > > **A3**: $\epsilon_d$ refers to the error $|V^*(b_d)-V^*_{\text{AdaOPS}}(b_d)|$ for any belief $b_d$ at depth $d$. The estimation error $\epsilon_{\text{est}}$ accounts for part of $\epsilon_d$ that does not arise from the descendant beliefs, i.e., $\epsilon_d=\epsilon_{\text{est}}+\gamma \epsilon_{d+1}$. For a not fused belief, $\epsilon_{\text{est}}$ is bounded by $\frac{\lambda}{1-\gamma}$. For a fused one, $\epsilon_{\text{est}}$ is at most $\frac{R_{\max}}{1-\gamma}\delta+\frac{\lambda}{1-\gamma}$. Hence, we can bound $\epsilon_{\text{est}}$ of any belief with $\frac{R_{\max}}{1-\gamma}\delta+\frac{\lambda}{1-\gamma}$.
> > >
> > > **Q4**: Does each inequality below line 110 hold with probability $1-\eta$ or do all such inequalities hold simultaneously with probability $1-\eta$?
> > >
> > > **A4**: All such inequalities hold simultaneously with probability $1-\eta$. It is possible to derive a step-by-step concentration following [Lim et al., 2020] addressing belief packing error at a finer level. However, this will further complicate the proof while bringing negligible improvement.
> > >
> > > Thank you very much for your advice. In the final version, we will clarify all the notions mentioned above and explicitly state all assumptions, making the proof more accessible.
> > >
> > > M. H. Lim, C. Tomlin, and Z. N. Sunberg, “Sparse Tree Search Optimality Guarantees in POMDPs with Continuous Observation Spaces,” in IJCAI, pp. 4135–4142, 2020.

---

### Official Review · Reviewer_abPZ · 2021-07-25

**Rating:** 7
**Confidence:** 4

**Summary:**

This paper presents an online method for approximately solving POMDPs, called AdaOPS. The method combines several known techniques for belief-tree construction and sampling. Specifically, it maintains upper and lower bound and uses the highest upper bound for action selection and weighted excess uncertainty for observation selection. It uses weighted particles to represent beliefs and use the weights together with KLD for resampling to improve belief estimates. It then uses belief packing to aggregate values of nearby beliefs. The methods are tested on four benchmark problems and indicate better performance compared to DESPOT and POMCPOW. Convergence results that expand the work in [18] are also presented.



**Limitations And Societal Impact:**

As any tools, methods for automated decision-making can be applied for both positive and negative purposes. This is akin to the fact that knife is very useful in our daily life, but of course, can be used for harm. The key is to ensure the application of the method is not used for negative purposes.

**Main Review:**

This paper presents an online method for approximately solving POMDPs, called AdaOPS. The method combines several known techniques for belief-tree construction and sampling. Specifically, it maintains upper and lower bound and uses the highest upper bound for action selection and weighted excess uncertainty for observation selection. It uses weighted particles to represent beliefs and use the weights together with KLD for resampling to improve belief estimates. It then uses belief packing to aggregate values of nearby beliefs. The methods are tested on four benchmark problems and indicate better performance compared to DESPOT and POMCPOW. Convergence results that expand the work in [18] are also presented.

I think the method is a relatively non-trivial combination of known components, though the components themselves and their applications to generating POMDP policies have been used in other solvers. In particular:
*/ The use of upper bound and weighted excess uncertainty are relatively common in POMDP solvers (as cited, [5, 14] have used them too).
*/ The use of KLD-sampling from particle filter[13]  to help identify the number of particles that need to be sampled in a sampled-based POMDP solver is new. Though, the idea of exploiting SIR techniques from particle filter to improve belief estimates without unnecessarily increasing the number of belief samples have been proposed in "M. Hoerger and H. Kurniawati. An On-Line POMDP Solver for Continuous Observation Spaces. ICRA 2021.".
*/ The idea of belief packing to aggregate values have also been applied in a POMDP solver since awhile back, albeit in an offline solver ("H. Kurniawati,  D. Hsu, and W. S. Lee. SARSOP: Efficient point-based POMDP planning by approximating optimally reachable belief spaces. RSS 2008.").

KLD-sampling can be used to incrementally increase the number of samples without requiring the effective sample size (ESS) nor posterior distribution to be computed. It would be useful to clarify why in the proposed method, there need to be a distinct step where ESS is used to determine whether to run KLD-sampling or not. The potential issue is that to keep computation fast, the ESS approximation used in the paper can actually be quite loose. Elaborating the trade-off between the two could be useful.


**Time Spent Reviewing:**

2.5

---

> ### Author Response · Authors · 2021-08-09
> **Response**
>
> Thanks a lot for your appreciation.
>
> In the final version, we will extend some discussion on the papers you mentioned. Besides, the LABECOP method proposed in [Hoerger et al., 2021] does not adopt the SIR particle filter in planning. Their method can be understood as using an SIS particle filter with increasing particle numbers for frequently visited action sequences.
>
> Below we address each of the raised questions/concerns.
>
> **Q1**: Why is there a distinct step where ESS is used to determine whether to run KLD-sampling or not?
>
> **A1**: KLD-Sampling process can indeed work independently without ESS. However, as pointed in line 77, frequent sampling causes sample impoverishment. Applying KLD-Sampling at each step, Equation (2) will, with the diminishing particle diversity, give decreasing particle numbers, which in turn aggravates the sample impoverishment. Therefore, it is beneficial to resample adaptively using ESS, ameliorating the sample impoverishment.
>
> **Q2**: Is there a trade-off between rapidness and accuracy in terms of the ESS approximation?
>
> **A2**: There is no trade-off on ESS approximation. The currently adopted ESS approximation is widely used and exhibits good empirical performance. In fact, it is unlikely to design a better one for the online planning scenario because, for a belief $b_t$, the integral $I$ we are interested in is
> $$
> I=\int \sum_{n=t}^{D-1}\gamma^{n-t}b_n(s_n)R(s_n,a_n)\mathrm{d}s_{t:D-1},
> $$
> where $s_{t:D-1}$ denotes the state sequence from timestep $t$ to $D-1$, and $D$ is the maximum searching depth. Notice that this integral is conditioned on future actions and observations that sum up to $|\mathcal{A}|(|\mathcal{A}||\mathcal{O}|)^{D-t-1}$. Thus, a reasonable ESS approximation should be general enough to accounts for all of them, rather than specific to a particular one, precluding a more refined approximation.

---

### Decision · Program_Chairs · 2021-09-27

**Decision:**

Accept (Poster)

**Comment:**

The paper describes a new online planning technique for POMDPs that introduces two new ideas: re-sampling based on KLD and belief packing.  The paper is well written and comprehensive in the sense that it includes a good empirical evaluation and a theoretical analysis.  The reviewers raised concerns about the scalability of the approach, the notation and the assumptions of the theoretical analysis.  The author response was very helpful in terms of clarifying those issues.  If the paper is accepted, the authors are strongly encouraged to follow the reviewers' advice.